# Towards Interpreting Deep Neural Networks via Understanding Layer Behaviors

## Abstract

Deep neural networks (DNNs) have achieved unprecedented practical success in many applications. However, how to interpret DNNs is still an open problem. In particular, what do hidden layers behave is not clearly understood. In this paper, relying on a teacher-student paradigm, we seek to understand the layer behaviors of DNNs by "monitoring" both across-layer and single-layer distribution evolution. Here, the "across-layer" and "single-layer" considers the layer behavior *along the depth* and a specific layer *along training epochs*, respectively. Relying on optimal transport theory, we employ the Wasserstein distance ($\mathcal{W}$-distance) to measure the divergence between the layer distribution and the target distribution. Theoretically, we prove that i) the $\mathcal{W}$-distance between the distribution of any layer and the target distribution tends to decrease along the depth. ii) For a specific layer, the $\mathcal{W}$-distance between the distribution in an iteration and the target distribution tends to decrease along training iterations. iii) However, a deep layer is not always better than a shallow layer for some samples. Moreover, our results helps to analyze the stability of layer distributions and explains why auxiliary losses help the training of DNNs. Extensive experiments justify our theoretical findings.

## 1 Introduction

Deep neural networks (DNNs) have been successfully applied in computer vision, such as image classification (Chen et al., 2019b; Hsu et al., 2019), image generation (Cao et al., 2019; Brock et al., 2019; Chrysos et al., 2019) and speech recognition (Yeh et al., 2019; Chen et al., 2019a). Despite their success, the internal mechanism of DNNs is still a black box. In particular, understanding what do hidden layers do and how to achieve remarkable performance remain persistently elusive. To answer these questions, we seek to understand across-layer and single-layer behavior.

For the across-layer behavior, we study the learning process by measuring the $\mathcal{W}$-distance between the distribution of any layer and the target distribution. Recently, most methods only focus on final predictions in different tasks (He et al., 2016). Due to the end-to-end training, interpreting each intermediate layer behaviors of a DNN, which, however, is still not clear. To provide interpretability, existing works try to produce a single prediction and observe the classification performance of each layer (Papernot & McDaniel, 2018; Szegedy et al., 2013; Kaya et al., 2019). For example, (Alain & Bengio, 2016) experimentally observe that the linear separability of features increases monotonically along the depth of a DNN. Unfortunately, there is no theoretical analysis to support the experimental finding. To understand the learning process of DNNs, one can explore the across-layer behavior by monitoring how the distributions propagate across different layers. However, how to open the internal mechanism of DNNs and investigate the across-layer behavior of a DNN remains an open question.

For the single-layer behavior, we seek to measure the $\mathcal{W}$-distance between the distribution in an iteration and the target distribution. It is important to analyze the distributional stability of one layer by measuring the change of the distributions. Recently, the distributional stability of a deep neural network attracts extensive attention (Santurkar et al., 2018). Some studies try to visualize the training behavior of one layer by plotting the mean and variance of features (Santurkar et al., 2018). Unfortunately, the visualizations are subjective and lack of necessary theoretical justifications. To this end, (Sonoda & Murata, 2019) propose a transport analysis method and state that a denoising Autoencoder transports mass to decrease the Shannon entropy of the data distribution. However, the

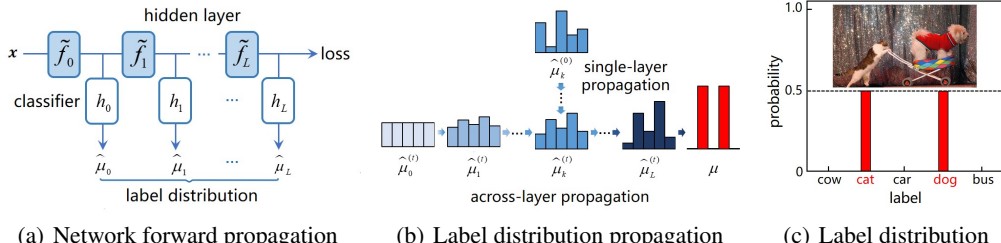

(a) Network forward propagation     (b) Label distribution propagation     (c) Label distribution

Figure 1: Demonstration of a DNN forward, distributions propagation and the intuition of the label distribution. (a) Given an input $\boldsymbol{x}$, we train hidden layers and classifiers to output label distributions $\widehat{\mu}_l$. (b) Across-layer propagation: the label distribution $\widehat{\mu}_0^{(t)}$ propagate to $\widehat{\mu}_L^{(t)}$. Single-layer propagation: the label distribution $\widehat{\mu}_k^{(0)}$ propagate to $\widehat{\mu}_k^{(t)}$ during the epochs in one layer. (c) Given an image with two labels (*i.e.*, cat and dog), the label distribution depicts their corresponding probabilities of 0.5.

method is limited and inflexible for analyzing a general case of deep neural networks. Therefore, it is very necessary and important to develop a new analytical method to interpret the stability of layers.

In this paper, we apply optimal transport theory to analyze the behavior of distributions. Specifically, we exploit Wasserstein divergence ($\mathcal{W}$-distance) to measure the difference between the distribution of any layer and the target distribution. By monitoring the change of the $\mathcal{W}$-distance, we are able to study both across-layer and single-layer behaviors. Our contributions are summarized as follows.

- We analyze the across-layer behavior and prove that the $\mathcal{W}$-distance between the distribution of any layer and the target distribution decreases along the depth of a DNN. It means that every layer of the network can express the target distribution progressively.
- We analyze the single-layer behavior and prove that for a specific layer, the $\mathcal{W}$-distance between the distribution in an iteration and the target distribution decreases across the training iterations when introducing a loss in the layer.
- Moreover, we provide experiments and theoretical justifications on these findings. The proposed analytical framework provides a different view of understanding and interpreting neural networks.

## 2 RELATED WORK

Many studies analyze the features of intermediate layers. (Bau et al., 2017) evaluates how hidden units align a set of semantic concepts to quantifies the interpretability of latent representations of a CNN. (Dosovitskiy & Brox, 2016) inverts image representations with up-convolutional networks for studying image representations. (Zeiler & Fergus, 2014) presents a visualization technique to investigate the function of intermediate feature layers and the classifier. (Zhang et al., 2018) learns an explanatory graph to reveal the knowledge hierarchy hidden inside a pre-trained CNN. (Zhang et al., 2019) learns a decision tree to clarify the specific reason for each semantic prediction of a CNN.

Some studies analyze the classification accuracy of a DNN. (Lee et al., 2015) introduces a classification objective to the individual hidden layers, in addition to the overall objective at the output layer. (Szegedy et al., 2015) uses auxiliary classifiers to improve the classification performance. (Kaya et al., 2019) introduces internal classifiers into off-the-shelf DNNs to understand overthinking of networks by studying how the prediction changes during a DNN's forward pass. (Gupta & Schütze, 2018) explains recurrent neural networks by understanding the layer-wise semantic accumulation behavior. (Sonoda & Murata, 2019) investigates the feature map inside a DNN by tracking the transport map, and prove that a deep Gaussian DAE transports mass to decrease Shannon entropy of the data distribution. (Alain & Bengio, 2016) experimentally observes that the linear separability of features increases monotonically along the depth of a DNN. However, there is no theoretical result to analyze this phenomenon. Existing studies using information bottleneck methods (Tishby & Zaslavsky, 2015; Bang et al., 2019) mainly analyze the dynamics of across different layers. However, it is hard for these methods to analyze the dynamics of a specific layer through different iterations. In contrast, our proposed method is able to analyze both single-layer and across-layer behaviors.

## 3 PROBLEM SETTING

**Notation.** We use bold lower-case letters (*e.g.*, $\boldsymbol{x}$) to denote vectors, and bold upper-case letters (*e.g.*, $\boldsymbol{X}$) to denote matrices. We denote the transpose of a vector (*e.g.*, $\boldsymbol{x}^{\mathsf{T}}$) or matrix (*e.g.*, $\boldsymbol{X}^{\mathsf{T}}$) by the superscript $^{\mathsf{T}}$. Let $\boldsymbol{1}:=[1,\ldots,1]^{\mathsf{T}}$ and $\mathbb{1}_{\mathcal{S}}(\cdot)$ be an indicator function of a set $\mathcal{S}$, and let $[n]=\{0,1,\ldots,n\}$. Let $*$ be the convolution operator, $\mathrm{grad}$ be gradient operator.

**Label distribution learning.** In this paper, we consider a label distribution learning problem (Geng, 2016), especially multi-label classification. In this setting, the label distribution is defined as a probability distribution to cover a certain number of labels, representing the degree to which each label describes the instance (see Figure 1 (c)). Note that the sum of the label distribution is equal to 1. Specifically, given training samples $\mathcal{S}$, we seek to learn a model $f$ from data space $\mathcal{X}$ into the label space $\mathcal{Y}$. In Figure 1(a), for $L$-layered neural networks, we denote $\tilde{f}_{0:l}=\tilde{f}_l\circ\cdots\circ\tilde{f}_0, l\leq L$ as the output of the $l$-th layers, then the label distribution mapping can be defined as $f_l:=h_l\circ\tilde{f}_{0:l}$, where $h_l$ is a probability function (*e.g.*, FC+softmax (Frogner et al., 2015)). Note that $f_l, l\in[L]$ have the same input domains and the same output domains. Our goal is to learn a model $f_l$ to let the predicted distribution $\widehat{\mu}_l$ close to the target distribution $\mu$. Then, we optimize the empirical risk:

$$\min_{f_l} \mathcal{L}(f_l) := \mathbb{E}_{\mathcal{S}}\left[d(\widehat{\mu}_l, \mu)\right], \quad \text{where} \quad \widehat{\mu}_l = f_{l\#}(\mu_0), \quad l \leq L, \tag{1}$$

where $\mathbb{E}_{\mathcal{S}}[\cdot]$ is the expectation *w.r.t.* the training set $\mathcal{S}$, $f_{l\#}$ denotes the pushforward operator of $\mu_0$, and $d(\cdot,\cdot)$ is some distribution divergence, such as Cross-entropy (CE) loss and Wasserstein loss. In practice, the label distribution is obtained by training with CE loss. Because of the convexity of CE loss, we derive the optimal label distribution for given features of every layer (Alain & Bengio, 2016). In this sense, the label distribution reflects the actual distribution of feature maps in a specific layer.

**Label distribution propagation.** In this paper, our goal is to explore how the layer distributions propagate in the across-layer and single-layer of a DNN. Specifically, we measure the divergence (*e.g.*, Wasserstein distance (Villani, 2008)) to observe how the layer distributions change across different layers or iterations. In Figure 1(b), we explain the behaviors of layer from the following two perspectives. For the across layers, measuring the distance between the distribution of any layer and the target distribution helps to understand the learning process and distribution expression ability. For a single layer, measuring the distance between the distribution in an training iteration and the target distribution helps to understand the dynamic behavior and training process of a DNN.

## 4 WASSERSTEIN DISTRIBUTION PROPAGATION QUANTIFICATION

**Optimal transport (OT).** Recently, the performance of multi-label classification can be improved by using $\mathcal{W}$-distance (Frogner et al., 2015; Zhao & Zhou, 2018). However, DNNs on the multi-label classification task still lack theoretical understanding. With the help of OT theory (Villani, 2008), $\mathcal{W}$-distance can be used to interpret DNNs. In contrast, some measures, such as Jensen–Shannon divergence, which, however, often ignore any metric structure of distribution (Frogner et al., 2015).

**Definition 1 ($\mathcal{W}$-distance (Villani, 2008))** *Given a prediction distribution $\widehat{\mu}$ and the target distribution $\mu$, and the cost matrix $\boldsymbol{C}$ defined as $C_{\kappa,\kappa'}=d_{\mathcal{K}}^p(\kappa,\kappa')$ with the metric $d_{\mathcal{K}}$, where $\kappa,\kappa'$ are label tags, then the $\mathcal{W}$-distance seeks to find a transportation matrix $\boldsymbol{T}$ by transporting a mass $\widehat{\mu}$ to $\mu$,*

$$\mathcal{W}^2(\widehat{\mu}, \mu) = \inf_{\boldsymbol{T}\in\Pi(\widehat{\mu},\mu)}\langle \boldsymbol{T}, \boldsymbol{C}\rangle, \tag{2}$$

*where $\Pi(\widehat{\mu},\mu)$ is the set of couplings and defined as $\Pi(\widehat{\mu},\mu)=\{\boldsymbol{T}\in\mathbb{R}_+^{K\times K}: \boldsymbol{T}\boldsymbol{1}=\widehat{\mu}, \boldsymbol{T}^{\mathsf{T}}\boldsymbol{1}=\mu\}$.*

In practice, the distance of two label tags in the cost matrix can be calculated by using word2vec (Mikolov et al., 2013) in the Euclidean space. We calculate $\mathcal{W}$-distance with entropic regularization by Sinkhorn algorithm (Knight, 2008) with $p=2$. See more details in Supplementary materials.

**Distribution propagation measure.** In this paper, we use $\mathcal{W}$-distance to measure the across-layer and single-layer label distribution propagation, as shown in Figure 1(b). For an across-layer, we measure $\mathcal{W}$-distance between the distribution of any layer and the target distribution, *i.e.*, $\mathcal{W}^2(\widehat{\mu}_1,\mu),\ldots,\mathcal{W}^2(\widehat{\mu}_L,\mu)$. In this way, we can measure how the distributions propagate across different layers. For the single-layer, we measure $\mathcal{W}$-distance between the distribution in an training iteration and the target distribution, *i.e.*, $\mathcal{W}^2(\widehat{\mu}_l^{(0)},\mu),\ldots,\mathcal{W}^2(\widehat{\mu}_l^{(t)},\mu)$. Therefore, we can study the behavior of distributions in the same layer.

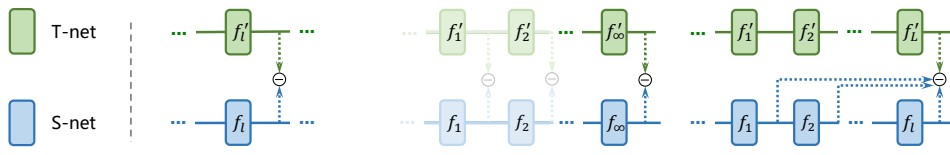

Figure 2: The teacher-student paradigm for analyzing layer behaviors of DNNs. We analyze one-layer behavior in (a), infinite-layer behavior in (b) and finite-layer behavior in (c).

## 5 TEACHER-STUDENT FRAMEWORK FOR LAYER BEHAVIOR ANALYSIS

Teacher-student analysis framework is widely used to analyze and understand DNNs (Tian, 2017; Du et al., 2018; Cuturi, 2013). For the one-layer behavior, this framework helps to understand the dynamic of student network from the teacher network. For the multi-layer behavior, this framework helps to study the ability of the student network to express distributions of the teacher network. Therefore, teacher-student analysis framework is important and necessary to understand DNNs.

### 5.1 ONE-LAYER BEHAVIOR ANALYSIS

In our problem setting, the target distribution in the training loss (1) can be represented by an output of a teacher network (T-net). Our goal is to make the output of the student network (S-net) to be close to the output of T-net, as shown in Figure 2 (a). Specifically, given a S-net $f$ and T-net $f'$, our goal is to find a function $f$ that minimizes the following training loss,

$$\mathcal{L}(f) := \mathbb{E}_{\mathcal{S}}\left[d(\widehat{\mu}_l, \mu)\right] := \mathbb{E}_{\boldsymbol{x} \sim \mathcal{S}}\left[\|f(\widetilde{\boldsymbol{x}}) - f'(\boldsymbol{x})\|^2\right], \tag{3}$$

where $\widetilde{\boldsymbol{x}} = \boldsymbol{x} + \boldsymbol{\varepsilon}$, $\boldsymbol{\varepsilon} \sim \nu_t$ is the corruption of data, and $\nu_t$ is a noise distribution *w.r.t.* the variance $t\boldsymbol{I}$, *i.e.*, $\nu_t = \mathcal{N}(0, t\boldsymbol{I})$. By optimizing the training loss, the distribution in one layer can be monitored by using a transport map. Next, we provide the definition of transport map below.

**Definition 2** (**Transport map**) *A transport map $f_t \colon \mathbb{R}^m \to \mathbb{R}^m$ can be defined as $f_t(\boldsymbol{x}) = \boldsymbol{x} + g_t(\boldsymbol{x})$ with an update direction $g_t$ when $t > 0$, and $f_0(\boldsymbol{x}) = \boldsymbol{x}$ when $t = 0$.*

Based on the definition of the transport map, we analyze the layer behavior of ResNet (He et al., 2016), *i.e.*, T-net is a residual block. Note that our analysis method can be extended to a more general case. Then, we solve the optimal transport map as follow.

**Theorem 1** (**Optimal transport map**) *If the teacher network is a residual unit $f'(\widetilde{\boldsymbol{x}}) = \boldsymbol{x} + \sigma(\widetilde{\boldsymbol{x}})$, where $\sigma$ contains FC layer and an activation function. When $\nu_t := \mathcal{N}(\boldsymbol{0}, t\boldsymbol{I})$, the minimum $f_t^*$ is*

$$f_t^*(\widetilde{\boldsymbol{x}}) = \widetilde{\boldsymbol{x}} + t\nabla \log(\nu_t * \mu_0)(\widetilde{\boldsymbol{x}}) + \sigma(\widetilde{\boldsymbol{x}}) := \widetilde{\boldsymbol{x}} + g_t(\widetilde{\boldsymbol{x}}).$$

The minimizer $f_t^*$ updates $\boldsymbol{x}$ along the update direction $g_t(\boldsymbol{x})$. Therefore, the optimal transport map $f_t^*$ with time $t$ transports the mass at the activation $\boldsymbol{x}$ toward $\boldsymbol{x} + g_t(\boldsymbol{x})$. To associate with continuity equation (Sonoda & Murata, 2019), we introduce Wasserstein gradient flow (WGF) as follows.

**Proposition 1** (**Wasserstein gradient flow**) *Assume $\mu_t$ satisfies the continuity equation with the gradient vector field $\partial_t \mu_t = -\nabla \cdot [\mu_t \nabla V_t]$ of a potential function $V_t$, and if the function $F$ satisfies $\partial_t F(\mu_t) = \int_{\mathbb{R}^m} -V_t(\boldsymbol{x})[\partial_t \mu_t](\boldsymbol{x})d\boldsymbol{x}$, then WGF coincides with $\partial_t \mu_t = -\mathrm{grad} F(\mu_t)$.*

Using the definition of WGF, we explore how the distribution propagate in one layer as follows.

**Theorem 2** *Based on the equivalent condition in Theorem 2 (Belavkin, 2016), and let $\Delta$ be the Laplacian operator. At the initial moment $t \to 0$, the pushforward $f_\# \mu_t$ with Gaussian distribution satisfies the backward heat equation (BHE), and evolves according to Wasserstein gradient flow:*

$$\partial_t \mu_{t=0}(\boldsymbol{x}) = -\Delta \mu_0(\boldsymbol{x}) = -\mathrm{grad}\,\mathcal{W}^2(\mu_t, \mu), \boldsymbol{x} \in \mathbb{R}^m.$$

Note that $\mathcal{W}$ is the potential function. Theorem 2 states that by introducing an auxiliary loss in a layer, the $\mathcal{W}$-distance can be decreased in the layer. It means that the label distribution can be close to the target distribution across training iterations.

## 5.2 Multi-layer Behavior Analysis

**(i) Infinitely deep neural networks.** For an infinitely DNN, we assume the number of layers and hidden units and the size of dataset is infinite. Specifically, let $0=t_0<t_1<\cdots<t_{L+1}=t$, and let $f_0$ be trained on $\mu_0$ with the variance $t_1-t_0$, then $\boldsymbol{x}_1:=f_0(\boldsymbol{x}_0)$ and its distribution is $\mu_1:=f_{0\#}\mu_0$. Then, we train $f_1$ on $\mu_1$ with the variance $t_2-t_1$. In the same way, we obtain $f_l(\boldsymbol{x}_l)$ and $\mu_l:=f_{l-1\#}\mu_{l-1}$. The composition of residual blocks can be written as $f_{0:L}^t(\boldsymbol{x}):=f_L\circ\cdots\circ f_0(\boldsymbol{x})$, where $t$ is total time. Then, the velocity of the composition coincides with the vector field $\partial_t f_{0:l}^{t=t_l}(\boldsymbol{x})=\nabla V_{t_l}(\boldsymbol{x})$.

Furthermore, we extend one-layer behavior analysis to the case of infinite layers, as shown in Figure 2 (b). Based on Theorem 1, we have the dynamical systems as follows.

**Lemma 1** *When the noise distribution is a Gaussian distribution, the continuous residual units is defined as a flow $\varphi_t\colon\mathbb{R}^m\to\mathbb{R}^m$ of the following dynamical systems associated with the vector field $\nabla V_t$, i.e., $\frac{d}{dt}\boldsymbol{x}(t)=\nabla\log(\mu_t(\boldsymbol{x}(t)))+\sigma(\boldsymbol{x}(t))$, where $t\geq 0$ and $\mu_t:=\varphi_{t\#}\mu_0$.*

Based on the proofs of Theorem 2, we can achieve a similar conclusion as follows.

**Theorem 3** *Based on the equivalent condition in Theorem 2 (Belavkin, 2016), and when the noise distribution is a Gaussian distribution, then the pushforward measure $\mu_t:=\varphi_{t\#}\mu_0$ evolves according to Wasserstein gradient flow as follows: $\frac{d}{dt}\mu_t(\boldsymbol{x})=-\mathrm{grad}\mathcal{W}^2[\mu_t,\mu], \mu_{t=0}=\mu_0$.*

Theorem 3 states that the $\mathcal{W}$-distance can be decreased across different layers. It means that the label distribution can be close to the target distribution along the depth of the layer.

**(ii) Finitely deep neural networks.** When the number of layers and the hidden unit number are finite, the above analysis is not available. As shown in Figure 2 (c), given a composition of $L$ T-nets, we try to analyze the ability of each layer of S-net to express the distribution of T-net. Hence, we derive an approximation bound for finitely deep neural networks. Here, Barron function is present a T-net and its definition is provided as follows.

**Definition 3** (**Barron function** (Sonoda & Murata, 2019)) *The function $\varphi$ is Barron function on $\mathcal{B}$ if $\varphi$ satisfies $\Omega_\mathcal{B}(C)=\{\varphi\colon\mathcal{B}\to\mathbb{R}\colon\exists\phi,\phi|_\mathcal{B}=\varphi,\|\phi\|_\mathcal{B}^*\leq C,\phi\in\mathcal{F}_\mathcal{B}\}$, $\|\phi\|_\mathcal{B}^*:=\int_{\mathbb{R}^m}\|\boldsymbol{w}\|_\mathcal{B}|\widehat{\phi}(\boldsymbol{w})|d\boldsymbol{w}$, where $\|\boldsymbol{w}\|_\mathcal{B}=\sup_{\boldsymbol{x}\in\mathcal{B}}|\langle\boldsymbol{w},\boldsymbol{x}\rangle|$ and $\widehat{\phi}$ is the Fourier transform of $\phi$, and $\mathcal{F}_\mathcal{B}$ is the set of functions with the Fourier inversion formula holds on $\mathcal{B}$.*

Based on the definition of Barron function, (Barron, 1993) states that a Barron function can be approximated by a neural network with one hidden layer. Given a fixed composition of $L$ Barron functions, we derive the error bound of the approximation between such composite function and a neural network with $l$ hidden layers. Note that $l$ is not necessarily equal to $L$. The approximation bound of *w.r.t.* Wasserstein distance can be provided as follows.

**Theorem 4** (**Wasserstein distribution approximation**) *Given a data distribution $\mu_0$ and a function $\varphi_i\colon\mathbb{R}^{m_{i-1}}\to\mathbb{R}^{m_i}$, and let $L_l=\log(l)+1$, if $\mathrm{support}(\mu_0)\subset\mathcal{K}_0$ and $\varphi_i(\mathcal{K}_{i-1})\subseteq\mathcal{K}_i, 1\leq i\leq l$, the function $\varphi_i$ is $\left(\frac{L_{l-1}-1}{L_l}\right)$-Lipschitz and is a Barron function, i.e., $\varphi_1\in\Omega_{\mathcal{K}_0}(C_0)$ and $\varphi_i\in\Omega_{\mathcal{K}_{i-1}+s\mathcal{B}_{m_{i-1}}}(C_i)$, then there exists a network $f$ with $l$ hidden layers with $\lceil 4C_l^2 m_l L_{l-1}^2 L_l^2/\epsilon^2\rceil$ neurons on the $i$-th layer,*

$$\mathcal{W}^2(\hat{\mu}_l,\mu)\leq\frac{\epsilon^2}{L_l^2}\left((2C_l\sqrt{m_l}+D_l)^2\frac{\delta}{s^2}+1\right),\quad l\leq L.\tag{4}$$

*where $D_l$ is the diameter of the set $\mathcal{K}_l$, and $\epsilon,\delta,s>0$ are parameters.*

Theorem 4 provides an error bound when using a neural network with $l$ hidden layers to approximate a fixed composition of $L$ Barron functions. Moreover, the upper bound decreases with the increasing of layers under certain conditions. Here, we can conclude that deep layers are not always better than shallow layer for some specific samples. Such samples are often classified correctly in the shallow layer rather than the deep layer, then the diameter $\mathcal{D}_l$ can be large for this set of samples. Hence, shallow layers can behave better than deep layers for these kinds of samples.

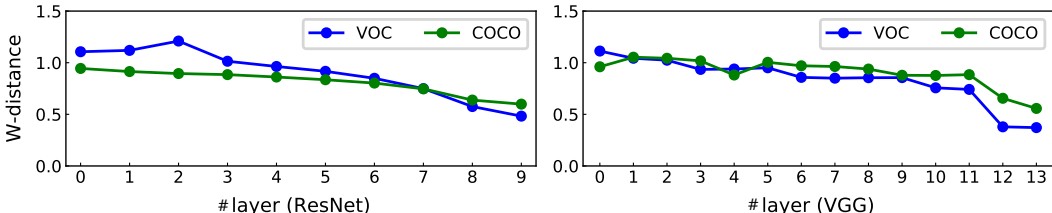

Figure 3: Wasserstein distance across different layers for different networks.

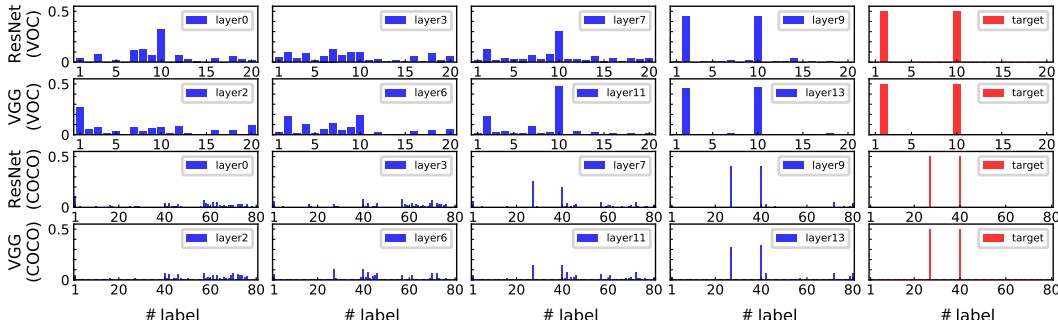

Figure 4: Distribution propagation across different layers for different networks.

## 6 EXPERIMENT

**Implementation Details.** All experiments are implemented on PyTorch. In the training, we use SGD optimizer with the initial learning rate as 0.01. The learning rate decays by a factor of 10 for every 40 epochs. The momentum and weight decay is set to be 0.9 and $10^{-4}$, respectively. We obtain models (including ResNet-18, ResNet-50 (He et al., 2016) and VGG-16 (Simonyan & Zisserman, 2014)) pre-trained on ImageNet(Deng et al., 2009) and train them for 100 epochs with the batch size of 128. Then, we freeze the model weights and train the internal classifiers (*i.e.*, the network $h_l$ in Figure 1(a)) for 100 epochs. Each internal classifier contains fully connected layer and Sigmoid function. We train the network using Binary Cross Entropy. We use training set to explore the distribution propagation of the across- and single-layer. In addition, we set $\alpha=0.01$ in Eqn. 17 to achieve the balanced results. For simplicity, we use ResNet to represent the ResNet-18 model and VGG to represent the VGG-16 model in the main paper.

**Data Sets and Evaluation Metrics.** We conduct experiments on two benchmark multi-label classification datasets, *i.e.*, VOC2007 (Everingham et al., 2010) and MS-COCO (Maas et al., 2013). VOC2007 (Everingham et al., 2010) contains 9,963 images from 20 object categories, which is divided into train, val and test sets. MS-COCO (Maas et al., 2013) contains 82,783 images as the training set and 40,504 images as the validation set. The objects are categorized into 80 classes with about 2.9 object labels per image. We use the classification accuracy, average per-class F1 (CF1), average overall F1 (OF1) and mean average precision (mAP) as the evaluation metrics.

### 6.1 ACROSS-LAYER BEHAVIOR EXPLORATION

#### 6.1.1 THE BEHAVIORS OF LABEL DISTRIBUTIONS IN DIFFERENT LAYERS

**How does every layer express the distribution?** Here, we study the expression ability of every layer, which is measured by Wasserstein distance ($\mathcal{W}$-distance) between the label distribution of any layer and the target distribution. As suggested in Theorem 4, deep layers have smaller error bound than shallow layers, meaning that they have better expression ability. From Figure 3, deep layers have smaller the $\mathcal{W}$-distance than shallow layers, which verifies the conclusion in Theorem 4. Moreover, the $\mathcal{W}$-distance decreases from shallow layers to deep layers. The shallow layers have similar ability to express the target distribution since they have similar values of the $\mathcal{W}$-distance. When approaching to the last layer, the $\mathcal{W}$-distance drops to a very small value. It implies that sufficient layers are able to express the target distribution. More results are shown in Section F in Supplementary materials.

Table 1: Improve performance of ResNet and VGG.

| Method | VOC | | | | COCO | | | |
|---|---|---|---|---|---|---|---|---|
| | accuracy | CF1 | OF1 | mAP | accuracy | CF1 | OF1 | mAP |
| ResNet | 64.48 | 58.02 | **59.10** | 85.18 | 27.33 | 55.65 | 60.30 | 64.36 |
| ResNet+EarlyExits | **66.01** | **58.67** | 58.76 | **85.49** | **30.03** | **57.49** | **61.38** | **67.28** |
| VGG | 68.96 | 58.58 | 59.71 | 88.48 | 32.41 | 59.37 | 62.94 | 70.64 |
| VGG-EarlyExits | **69.85** | **59.49** | **59.94** | **88.57** | **33.95** | **60.32** | **63.73** | **71.95** |

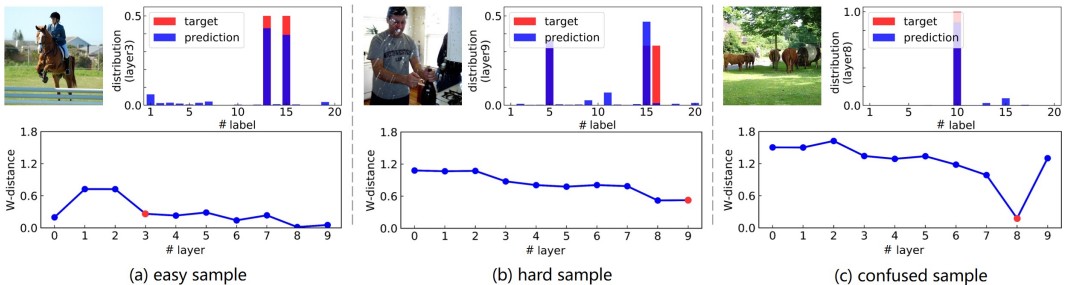

(a) easy sample  (b) hard sample  (c) confused sample

Figure 5: Demonstration of properties on easy, hard and confused samples.

**How does the distribution propagate across layers?** In this experiment, we investigate how the label distribution propagate across layers to understand the learning process from layer to layer in a DNN. Note that we normalize the prediction of classifier as the probability distribution (*i.e.*, the sum of label prediction probabilities of one sample is 1.). Figure 4 shows that the label distribution of one sample propagates from the first layer to the last layer. In the shallow layers, the label distribution is far away from the target distribution, then it can be close to the target distribution in the deep layers. Different from the decreasing tendency in Figure 3, the label distribution of one sample may not close to the target distribution progressively. For example, in the first row of Figure 4, the probability of the 10-th class is large in the 0-th layer, but goes down in the 3-th layer.

### 6.1.2 THE EVALUATION OF ACROSS-LAYER'S BEHAVIORS

**How to evaluate the quality of each layer?** Existing methods attempt to visualize the feature maps to evaluate the quality of each layer. However, observing the feature maps is difficult because of its high dimensionality and complexity. Relying on the $\mathcal{W}$-distance, we can evaluate the quality of each layer by measuring the ability of expression. If the $\mathcal{W}$-distance decreases rapidly in a layer, it suggests that the quality of this layer is good. Taking ResNet on VOC in Figure 3 as an example, $0{\sim}5$ layers have similar value of the $\mathcal{W}$-distance, while $6{\sim}8$ layers have quite different value of the $\mathcal{W}$-distance. It means that these deep layers (*i.e.*, $6{\sim}8$) are better than the shallow layers (*i.e.*, $0{\sim}5$).

### 6.1.3 HOW TO EXPLOIT THE ACROSS-LAYER BEHAVIORS

**Does every sample contribute equally?** In practice, a deep neural network construct more complex features progressively throughout the layers (Lee et al., 2011). An interesting question arises: does the contribution of each sample vary across different layers and training iterations? To answer this question, we first define the concept of sample difficulty in terms of the $\mathcal{W}$-distance. As shown in Figure 5, the samples can be divided into three categories, including easy, hard and confused. The intuition is that: 1) Easy samples should have small the $\mathcal{W}$-distance in the first few layers, *i.e.*, , they should be classified correctly with high confidence in a shallow layer. 2) Hard samples would have large the $\mathcal{W}$-distance in a deeper layer, *i.e.*, , it cannot be resolved at all, or only near the last layer. 3) Confused samples have small the $\mathcal{W}$-distance in a shallow layer, while have large the $\mathcal{W}$-distance in a deep layer. It means that although these samples can be classified correctly in the shallow layer, they still become misclassified at the last layer. Considering the difficulty of samples would help to interpret the training of models and design the training loss, which may speed up convergence and improve the performance eventually. We show more results in Section I in Supplementary materials.

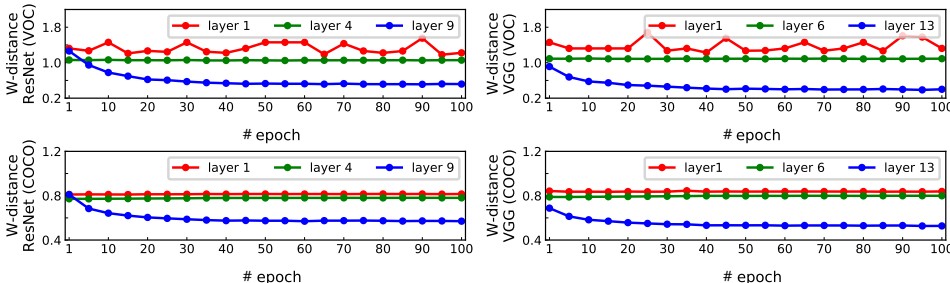

Figure 6: Wasserstein distance between the distribution in an epoch and the target distribution across different training epochs for different networks. We choose the $1, 4, 9$-th layer for ResNet and the $1, 6, 13$-layer for VGG.

**How to improve classification performance?** We will discuss how to exploit the behaviors of different layers to improve the model performance. In practice, we observe that in deep neural networks, some samples are correctly classified in the intermediate layers but misclassified in the last layer. For example, 35.52% and 31.04% of the samples for ResNet and VGG are misclassified on the test set of VOC2007, respectively. For these samples, 5.96% and 7.76% samples are correctly classified in all intermediate layers of ResNet and VGG, respectively. Based on this phenomenon, we can improve the performance by early exiting such confused samples. Different from the strategy of SDN (Kaya et al., 2019), how to early exit for multi-label classification DNNs is very challenging. Therefore, we design a new early-exits strategy for multi-label prediction. In Table 1, our proposed method consistently outperforms the baseline methods. We give more details about early-exits strategy and show more results in Section H in Supplementary materials.

## 6.2 SINGLE-LAYER BEHAVIOR EXPLORATION

In this section, we study how the distribution of an intermediate layer change during the training process. We consider to use $\mathcal{W}$-distance to measure the stability of distribution. We propose theoretical and empirical analysis for the distribution stability in one layer.

**How the distribution propagate during updating DNNs?** In this experiment, we investigate how distributions propagate when training DNNs. From Figure 6, the label distribution in the first layers of ResNet or VGG often fluctuates significantly due to the limited discriminative power of very shallow layers. When approaching the last layer, the supervision is sufficient to decrease the $\mathcal{W}$-distance. These experiments justify Theorem 2 that the $\mathcal{W}$-distance can be decreased with sufficient supervision. More results are shown in Section G in Supplementary materials.

**How stable is the distribution in one layer?** The training stability is a key problem in DNNs and can be largely addressed by the widely used Batch Normalization (BN) (Ioffe & Szegedy, 2015). However, we prove that the stability of distribution is hard to be guaranteed even with BN (See proof in supplementary). In this paper, we use the $\mathcal{W}$-distance to measure the discrepancy between the distribution in an iteration and the target distribution across different training iterations in the specific layer. Motivated by this, we further measure stability of label distribution during training DNNs. The conventional understanding of BN suggests that the $\mathcal{W}$-distance should decrease. Note that, from Figure 6, the first layer of ResNet or VGG may fluctuate during the training process, which indicates that deep layers often have better ability than shallow layers to express the target distribution.

## 7 CONCLUSION

In this paper, we have interpreted deep neural networks (DNNs) via understanding layer behaviors. With the help of optimal transport theory, we propose a unified teacher-student analysis framework to investigate the across-layer and single-layer behaviors of a DNN. Theoretically, we prove that i) the $\mathcal{W}$-distance between the distribution of any layer and the target distribution decreases along the depth. ii) the $\mathcal{W}$-distance of a specific layer between the distribution in an iteration and the target distribution decreases along training iterations. Extensive experiments justify our theoretical findings. The proposed analytical framework can facilitate future researches to interpret a deep neural network.

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

# SUPPLEMENTARY ON "TOWARDS INTERPRETING DEEP NEURAL NETWORKS VIA UNDERSTANDING LAYER BEHAVIORS"

## A PRELIMINARY

**Notation.** Specifically, we use bold lower-case letters (*e.g.*, $\boldsymbol{x}$) to denote vectors, and bold upper-case letters (*e.g.*, $\boldsymbol{X}$) to denote matrices. We denote the transpose of a vector (*e.g.*, $\boldsymbol{x}^\mathsf{T}$) or matrix (*e.g.*, $\boldsymbol{X}^\mathsf{T}$) by the superscript $^\mathsf{T}$. Let $\boldsymbol{1} := [1, \ldots, 1]^T$. For two matrices $\boldsymbol{A}$ and $\boldsymbol{B}$ of the same size, their inner-product is $\langle \boldsymbol{A}, \boldsymbol{B} \rangle = \mathrm{tr}(\boldsymbol{A}^\mathsf{T} \boldsymbol{B})$. Let $*$ be the convolution operator. Let $\| \cdot \| = \| \cdot \|_2$ denote Euclidean norm on vectors in $\mathbb{R}^n$. For a function $f$, let $f^\vee := f(-\boldsymbol{x})$. Let $\Delta f = \sum_{i=1}^n \frac{\partial^2 f}{\partial^2 \boldsymbol{x}_i}$ denote the Laplacian. Let $\mathcal{B}_n$ be the unit ball in $n$ dimension, *i.e.*, $\mathcal{B}_n = \{ \boldsymbol{x} \in \mathbb{R}^n : \|\boldsymbol{x}\| \leq 1 \}$. For two sets $\mathcal{A}$ and $\mathcal{B}$ and a scalar $r \in \mathbb{R}$, let $\mathcal{A} + \mathcal{B} = \{ \boldsymbol{x} + \boldsymbol{y} : \boldsymbol{x} \in \mathcal{A}, \boldsymbol{y} \in \mathcal{B} \}$ and $r\mathcal{A} = \{ r\boldsymbol{x} : \boldsymbol{x} \in \mathcal{A} \}$.

**Definition 4** (**Lipschitz continuous**) *A function $\varphi \colon \mathbb{R}^m \to \mathbb{R}^n$ is called L-Lipschitz continuous if there exists a real constant $L \geq 0$ such that, for all $\boldsymbol{x}$ and $\boldsymbol{y}$ w.r.t. the $L^2$ norm,*

$$\|\varphi(\boldsymbol{x}) - \varphi(\boldsymbol{y})\|_2 \leq L \|\boldsymbol{x} - \boldsymbol{y}\|_2.$$

**Definition 5** (**Fourier transform**) *For $\varphi \in L^1(\mathbb{R})$, Fourier transform of $\varphi \colon \mathbb{R}^n \to \mathbb{R}$ is defined as:*

$$\widehat{\varphi}(\boldsymbol{w}) := \frac{1}{(2\pi)^{\frac{n}{2}}} \int_{\mathbb{R}^n} \varphi(\boldsymbol{x}) e^{-i\langle \boldsymbol{w}, \boldsymbol{x} \rangle} d\boldsymbol{x}. \tag{5}$$

For vector-valued functions $\varphi \colon \mathbb{R}^n \to \mathbb{R}^m$, we conduct component-wise Fourier transform.
Let $\widehat{\varphi}^\vee(\boldsymbol{x}) := \widehat{\varphi}(-\boldsymbol{x})$, the inverse Fourier transform is

$$(\mathcal{F}^{-1}\varphi)(\boldsymbol{x}) := \int_{\mathbb{R}^n} \varphi(\boldsymbol{w}) e^{i\langle \boldsymbol{w}, \boldsymbol{x} \rangle} d\boldsymbol{x} = (2\pi)^{\frac{n}{2}} \widehat{\varphi}^\vee. \tag{6}$$

**Definition 6** *For a bounded set $\mathcal{B}$, the norm of a function $f \colon \mathbb{R}^n \to \mathbb{R}$ can be defined on a set $\mathcal{B}$, $\|f\|_{\mathcal{B}}^* := \int_{\mathbb{R}^n} \|\boldsymbol{w}\|_{\mathcal{B}} |\hat{f}(\boldsymbol{w})| d\boldsymbol{w}$, where $\|\boldsymbol{w}\|_{\mathcal{B}} = \sup_{\boldsymbol{x} \in \mathcal{B}} |\langle \boldsymbol{w}, \boldsymbol{x} \rangle|$.*

## B PROOF OF THEOREM 4

**Definition 7** (**Barron function** (Sonoda & Murata, 2019)) *The function $\varphi$ is Barron function on $\mathcal{B}$ if $\varphi$ satisfies $\Omega_{\mathcal{B}}(C) = \{ \varphi \colon \mathcal{B} \to \mathbb{R} \colon \exists \phi, \phi|_{\mathcal{B}} = \varphi, \|\phi\|_{\mathcal{B}}^* \leq C, \phi \in \mathcal{F}_{\mathcal{B}} \}$, where $\|\phi\|_{\mathcal{B}}^* := \int_{\mathbb{R}^m} \|\boldsymbol{w}\|_{\mathcal{B}} |\widehat{\phi}(\boldsymbol{w})| d\boldsymbol{w}$, where $\|\boldsymbol{w}\|_{\mathcal{B}} = \sup_{\boldsymbol{x} \in \mathcal{B}} |\langle \boldsymbol{w}, \boldsymbol{x} \rangle|$ and $\widehat{\phi}$ is the Fourier transform of $\phi$, and $\mathcal{F}_{\mathcal{B}}$ is the set of functions with the Fourier inversion formula holds on $\mathcal{B}$,i.e.,*

$$\mathcal{F}_{\mathcal{B}} = \left\{ \phi \colon \mathbb{R}^m \to \mathbb{R} \colon \forall \boldsymbol{x} \in \mathcal{B}, \phi(\boldsymbol{x}) = \phi(\boldsymbol{0}) + \int \left( e^{i\langle \boldsymbol{w}, \boldsymbol{x} \rangle} - 1 \right) \widehat{\phi}(\boldsymbol{w}) d\boldsymbol{w} \right\}. \tag{7}$$

**Definition 8** (**Sigmoidal function**) *A sigmoidal function is a bounded measurable function $\sigma \colon \mathbb{R} \to \mathbb{R}$ such that $\lim_{x \to -\infty} \sigma(x) = 0$ and $\lim_{x \to +\infty} \sigma(x) = 1$.*

**Theorem 5** (**Barron theorem (Barron, 1993)**) *Let $\mathcal{B} \subseteq \mathbb{R}^d$ be a bounded set, and $\mu$ be any probability measure on $\mathcal{B}$, $\varphi \in \Omega_{\mathcal{B}}(C)$ and $f(\boldsymbol{x}) = \sum_{i=1}^n c_i \sigma(\langle \boldsymbol{w}_i, \boldsymbol{x} \rangle + b_i)$, where $\sigma(\cdot)$ be a sigmoidal function, then there exist $\boldsymbol{w}_i \in \mathbb{R}^d, b_i \in \mathbb{R}, c_i \in \mathbb{R}$ with $\sum_{i=1}^n |c_i| \leq 2C$ such that*

$$\|\varphi(\boldsymbol{x}) - f(\boldsymbol{x})\|_\mu^2 := \int_{\mathcal{B}} (\varphi(\boldsymbol{x}) - f(\boldsymbol{x}))^2 \mu(d\boldsymbol{x}) \leq \frac{(2C)^2}{n}. \tag{8}$$

We extend the above theorem to the case of the composition of Barron functions denoted as $\varphi_{j:i} := \varphi_j \circ \varphi_{j-1} \circ \cdots \circ \varphi_i$.

**Corollary 1** *Let $\mathcal{B}$ be a bounded set, $\mu$ be any probability measure and the neural network $f(\boldsymbol{x}) = \sum_{i=1}^n c_i \sigma(\langle \boldsymbol{w}_i, \boldsymbol{x} \rangle + b_i)$, where $\sigma(\cdot)$ is a sigmoidal function. If the composition of Barron*

functions is restricted in the set of $\Omega_{\mathcal{B}}(C)$, then there exists $\boldsymbol{w}_i \in \mathbb{R}^d, b_i \in \mathbb{R}, c_i \in \mathbb{R}$ with $\sum_{i=1}^n |c_i| \leq 2C$ such that

$$\|\varphi_{k:1}(\boldsymbol{x}) - f(\boldsymbol{x})\|_\mu^2 := \int_{\mathcal{B}} (\varphi_{k:1}(\boldsymbol{x}) - f(\boldsymbol{x}))^2 \, \mu(d\boldsymbol{x}) \leq \frac{(2C)^2}{n}. \tag{9}$$

**Proof**   Directly using Lemma 1 and Theorem 2 of (Barron, 1993) can complete the conclusion. □

**Theorem 6 (Extend Barron theorem)** *Let $\mathcal{B}$ be a bounded set, $\mu$ be any probability measure and $f(\boldsymbol{x}) = \sum_{i=1}^n c_i \sigma(\langle \boldsymbol{w}_i, \boldsymbol{x} \rangle + b_i)$, where $\sigma(\cdot)$ be a sigmoidal function. If* support$(\mu_0) \subset \mathcal{K}_0$ *and $\varphi_i(\mathcal{K}_{i-1}) \subseteq \mathcal{K}_i, 1 \leq i \leq l$, the function $\varphi_i$ is $\frac{\log(l-1)}{\log(l)+1}$-Lipschitz and Barron function, i.e., $\varphi_1 \in \Omega_{\mathcal{K}_0}(C_0)$ and $\varphi_i \in \Omega_{\mathcal{K}_{i-1}+s\mathcal{B}_{m_{i-1}}}(C_i)$, then there exists a neural network $f$ with $l$ hidden layers and $\mathcal{S} \subset \mathbb{R}^{d_0}$ satisfying $\mu_0(\mathcal{S}) \leq 1 - \frac{\epsilon^2}{s^2(\log(l-1)+1)^2}$ so that*

$$\left( \int \mathbb{1}_{\mathcal{S}} \|\varphi - f\|^2 d\mu_0 \right)^{\frac{1}{2}} \leq \frac{\epsilon}{\log(l)+1}. \tag{10}$$

**Proof**   For simplicity, let $\varphi := \varphi_{l:1} = \varphi_l \circ \cdots \circ \varphi_1$ be the composition of Barron functions, especially $\varphi_1 = \widetilde{\varphi}_{k:1}$, where $\widetilde{\varphi}_l$ is defined in Corollary 1. Let $f := f_{l:1} = f_l \circ f_{l-1} \circ \cdots \circ f_1$ be the composition of $k$-layered neural network, where $f_i : \mathbb{R}^{d_{i-1}} \to \mathbb{R}^{d_{i-1}}$ are functions defined as

$$(f_i(\boldsymbol{x}))_j = \sum_{k=1}^{r_i} c_{ijk} \cdot \sigma(\langle \boldsymbol{w}_{ijk}, \boldsymbol{x} \rangle + b_{ijk}), \tag{11}$$

where $c_{ijk}, b_{ijk} \in \mathbb{R}$ and $\boldsymbol{w}_{ijk} \in \mathbb{R}^{d_{i-1}}$ are parameters, and $r_i$ is the number of nodes in the $i$-th layer. Note that the function $f_i$ is represented by a neural network with one hidden layer and a linear output layer. We prove the theorem by induction on $l$.

i) For $l=1$, using assumptions of the support of initial distribution (*i.e.*, support$(\mu_0) \subset \mathcal{K}_0$) and the definition of Barron function. Then, $\epsilon = (2C_1)^2 / r_1$ and $\sum_{i=1}^{r_1} |c_i| \leq 2C_1$, we complete the results.

ii) For $l>1$, by the induction step, we assume the functions $f_1, \ldots, f_{l-1}$ satisfy the conclusion for $\widetilde{\varphi}_1, \ldots, \widetilde{\varphi}_{l-1}$. Let $\mathcal{S}_{l-1}$ be the set in the conclusion. Applying Corollary 1 to $f_l$ to get that for each $1 \leq j \leq m_j$, for any $\mu$ supported on a set $K'_{l-1} \subseteq \mathbb{R}^{m_{l-1}}$ and any $r_l \in \mathbb{N}$, there exists a neural net $f_{l,j}$ with 1 hidden layer with $r_l$ nodes such that

$$\left( \int_{\mathbb{R}^{m_{l-1}}} (\varphi_{l,j} - f_{l,j})^2 d\mu \right)^{\frac{1}{2}} \leq \frac{2C_{f_l, \mathcal{K}'_{l-1}}}{\sqrt{r_l}}.$$

Note that Theorem 1 applies to any distribution $\mu$ supported on the set $\mathcal{K}'_{l-1}$. Let $\mathcal{S}_l = S_{l-1} \cap \{\boldsymbol{x} : f_{l-1:1}(\boldsymbol{x}) \in \mathcal{K}_{l-1} + s\mathcal{B}_{m_{l-1}}\}$. Applying Theorem 1 with $\mathcal{K}'_l = \mathcal{K}_l + s\mathcal{B}_{m_l}, r_l = \left\lceil \frac{4C_l^2 m_l (\log(l-1)+1)^2 (\log(l)+1)^2}{\epsilon^2} \right\rceil$ and $\mu = f_{l-1:1\#}(\mathbb{1}_{\mathcal{S}_l} \mu_0)$. We have that $\mu$ is supported on $g_{l-1:1}(\mathcal{S}_l) \subseteq \mathcal{K}_{l-1} + s\mathcal{B}_{m_{l-1}} = \mathcal{K}'_{l-1}$, and the function $\varphi_l$ is $C_l$-Barron function on this set by the assumption. The conclusion of Theorem 1 gives $(f_l)_j$ such that

$$\left( \int_{\mathbb{R}^{m_{l-1}}} (\varphi_{l,j} - f_{l,j})^2 d(f_{l-1:1\#}(\mathbb{1}_{\mathcal{S}_l} \mu_0)) \right)^{\frac{1}{2}} \leq \frac{2C_l}{\sqrt{r_l}} \leq \frac{\epsilon}{\sqrt{m_l}(\log(l-1)+1)(\log(l)+1)}.$$

For all elements of $\varphi_l$ and $f_l$, we have

$$\left( \int_{\mathbb{R}^{m_{l-1}}} \|\varphi_l - f_l\|^2 d(f_{l-1:1\#}(\mathbb{1}_{\mathcal{S}_l} \mu_0)) \right)^{\frac{1}{2}} \leq \frac{\epsilon}{(\log(l-1)+1)(\log(l)+1)}.$$

We bound by the triangle inequality

$$\left( \int_{\mathbb{R}^m} \mathbb{1}_{\mathcal{S}_l} \|\varphi_{l:1} - f_{l:1}\|^2 d\mu_0 \right)^{\frac{1}{2}}$$

$$\leq \left( \int_{\mathbb{R}^m} \mathbb{1}_{\mathcal{S}_l} \|\varphi_l \circ \varphi_{l-1:1} - \varphi_l \circ f_{l-1:1}\|^2 d\mu_0 \right)^{\frac{1}{2}} + \left( \int_{\mathbb{R}^m} \mathbb{1}_{\mathcal{S}_l} \|\varphi_l \circ f_{l-1:1} - f_l \circ f_{l-1:1}\|^2 d\mu_0 \right)^{\frac{1}{2}}$$

$$\leq \left( \int_{\mathbb{R}^m} \mathbb{1}_{\mathcal{S}_l} \|\varphi_l \circ \varphi_{l-1:1} - \varphi_l \circ f_{l-1:1}\|^2 d\mu_0 \right)^{\frac{1}{2}} + \left( \int_{\mathbb{R}^m} \mathbb{1}_{\mathcal{S}_l} \|\varphi_l - f_l\|^2 df_{l-1:1\#}(\mathbb{1}_{\mathcal{S}_l}\mu_0) \right)^{\frac{1}{2}}$$

$$\leq \mathrm{Lip}(\varphi_l) \left( \int_{\mathbb{R}^m} \mathbb{1}_{\mathcal{S}_l} \|\varphi_{l-1:1} - f_{l-1:1}\|^2 d\mu_0 \right)^{\frac{1}{2}} + \frac{\epsilon}{(\log(l-1)+1)(\log(l)+1)}$$

$$\leq \mathrm{Lip}(\varphi_l) \left( \int_{\mathbb{R}^m} \mathbb{1}_{\mathcal{S}_{l-1}} \|\varphi_{l-1:1} - f_{l-1:1}\|^2 d\mu_0 \right)^{\frac{1}{2}} + \frac{\epsilon}{(\log(l-1)+1)(\log(l)+1)}$$

$$\leq \frac{\log(l-1)}{\log(l)+1} \cdot \frac{\epsilon}{\log(l-1)+1} + \frac{\epsilon}{(\log(l-1)+1)(\log(l)+1)}$$

$$\leq \frac{\epsilon}{\log(l)+1},$$

where the last inequality holds by the assumption (*i.e.*, $\mathrm{Lip}(\varphi_l) = \frac{\log(l-1)}{log(l)+1}$) and the induction hypothesis.

As above analysis, we have

$$\int \mathbb{1}_{\mathcal{S}_{l-1}} \|\varphi_{l-1:1} - f_{l-1:1} d\mu_0\|^2 d\mu_0 \leq \frac{\epsilon^2}{(\log(l-1)+1)^2},$$

by the induction hypothesis. Also, $\varphi_{l-1:1} \in \mathcal{K}_{l-1}$ for all $\boldsymbol{x} \in \mathrm{support}(\mu_0)$ by the assumption of the theorem. Therefore, by Markov's inequality and the induction hypothesis on $\mathcal{S}_{l-1}$,

$$\mu_0 \left( \mathcal{S}_{l-1} \cap \left\{ \boldsymbol{x} : \boldsymbol{x} \notin \mathcal{K}_{l-1} + s f_{l-1:1}(\mathcal{B}_{m_{l-1}}) \right\} \right)$$

$$\leq \mu_0 \left( \mathcal{S}_{l-1} \cap \{ \boldsymbol{x} : \|\varphi_{l-1:1} - f_{l-1:1}\| \geq s \} \right) \leq \frac{\epsilon^2}{s^2 (\log(l-1)+1)^2}.$$

Therefore,

$$\mu_0(\mathcal{S}_l) \leq \mu_0(\mathcal{S}_{l-1}) - \frac{\epsilon^2}{s^2 (\log(l-1)+1)^2},$$

the last equality holds by a fact that there exists a constant $\delta$ to satisfy $\mu_0(\mathcal{S}_l) = 1 - \frac{\delta \epsilon^2}{s^2 (\log(l-1)+1)^2}$. $\square$

Based on the above conclusions, we prove Theorem 4 as follow.

**Theorem 4** *Given a data distribution $\mu_0$ and a function $\varphi_i \colon \mathbb{R}^{m_{i-1}} \to \mathbb{R}^{m_i}$, and let $L_l = \log(l)+1$, if $\mathrm{support}(\mu_0) \subset \mathcal{K}_0$ and $\varphi_i(\mathcal{K}_{i-1}) \subseteq \mathcal{K}_i$, $1 \leq i \leq l$, the function $\varphi_i$ is $\left( \frac{L_{l-1}-1}{L_l} \right)$-Lipschitz and is a Barron function, i.e., $\varphi_1 \in \Omega_{\mathcal{K}_0}(C_0)$ and $\varphi_i \in \Omega_{\mathcal{K}_{i-1}+s\mathcal{B}_{m_{i-1}}}(C_i)$, then there exists a network $f$ with $l$ hidden layers with $\lceil 4 C_l^2 m_l L_{l-1}^2 L_l^2 / \epsilon^2 \rceil$ neurons on the $i$-th layer,*

$$\mathcal{W}^2(\hat{\mu}_l, \mu) \leq \frac{\epsilon^2}{L_l^2} \left( (2 C_l \sqrt{m_l} + D_l)^2 \frac{\delta}{s^2} + 1 \right), \quad l \leq L. \tag{12}$$

*where $D_l$ is the diameter of the set $\mathcal{K}_l$, and $\epsilon, \delta, s > 0$ are parameters.*

**Proof** The functions $f_1, \cdots, f_l$ in Theorem 6 satisfy that

$$\int \mathbb{1}_{\mathcal{S}_{l-1}} \|\varphi_{l-1:1} - f_{l-1:1} d\mu_0\|^2 d\mu_0 \leq \frac{\epsilon^2}{(\log(l-1)+1)^2}.$$

The range of $f_l = ((f_l)_1, \cdots, (f_l)_{m_l})$ is contained in a set of diameter $2 C_l \sqrt{m_l}$ because the activation in a neural network is ranged in $[0, 1]$ and the weights $c_{ijk}$ in Theorem 6 satisfy $\sum_{k=1}^r |c_{ijk}| \leq 2 C_l$.

Choose a constant vector $k$ to minimize $\int_{\mathcal{S}_l} \|\varphi_{l:1}(\boldsymbol{x}) - f_{l:1} - k\|^2 d\mu_0$ and replace $f_l$ with $f_l + k$. Note that the range of $f_l$ and $\varphi_l$ necessarily overlap. We still have $\int_{\mathcal{S}_l} \|\varphi_{l:1}(\boldsymbol{x}) - f_{l:1}\|^2 d\mu_0 \leq \frac{\epsilon^2}{(\log(l)+1)^2}$. Moreover, we have

$$\|\varphi_l(\boldsymbol{x}) - f_l(\boldsymbol{x})\| \leq 2C_l\sqrt{m_l} + D_l,$$

for $\boldsymbol{x} \in \mathcal{K}_0$, where $D_l$ is the diameter of $\mathcal{K}_l$. Let $\mathcal{S}_l^c$ be a complementary of $\mathcal{S}_l$, we have

$$\mu(\mathcal{S}_l^c) = \frac{\delta\epsilon^2}{s^2(\log(l-1)+1)^2},$$

then we have

$$\int_{\mathcal{K}_0} \|\varphi_{l:1} - f_{l:1}\|^2 d\mu_0 \leq \int_{\mathcal{S}_l} \|\varphi_{l:1} - f_{l:1}\|^2 d\mu_0 + \int_{\mathcal{S}_l^c} \|\varphi_{l:1} - f_{l:1}\|^2 d\mu_0$$

$$\leq \frac{\epsilon^2}{(\log(l)+1)^2} + (2C_l\sqrt{m_l} + D_l)^2 \frac{\delta\epsilon^2}{s^2(\log(l-1)+1)^2}$$

$$= \frac{\epsilon^2}{(\log(l)+1)^2} \left( (2C_l\sqrt{m_l} + D_l)^2 \frac{\delta}{s^2} + 1 \right).$$

For $(\varphi_{l:1}(X), f_{l:1}(X)), X \sim \mu_0$ define a coupling between the distributions. Based on the definition of $\mathcal{W}$-distance, we have

$$\mathcal{W}^2(\hat{\mu}_k, \mu) = \mathbb{E}_{X \sim \mu_0} \|\varphi_{l:1} - f_{l:1}\|^2$$

$$= \int_{\mathcal{K}_0} \|\varphi_{l:1} - f_{l:1}\|^2 d\mu_0$$

$$\leq \frac{\epsilon^2}{(\log(l)+1)^2} \left( (2C_l\sqrt{m_l} + D_l)^2 \frac{\delta}{s^2} + 1 \right).$$

$\square$

## C   PROOFS OF LAYER BEHAVIOR ANALYSIS OF DEEP NEURAL NETWORKS

**Definition 9** *(Flow) A flow $\varphi_t$ is given by an ordinary differential equation (ODE), $\dot{\varphi}_t(\boldsymbol{x}) = \boldsymbol{v}_t(\varphi_t(\boldsymbol{x}))$ with a velocity field $\boldsymbol{v}_t$ when $t > 0$, and $\varphi_0(\boldsymbol{x}) = \boldsymbol{x}$ when $t = 0$.*

**Proposition 2** **(Continuity equation** (Sonoda & Murata, 2019)) *Let $\varphi_t$ be the flow of an ODE with vector field $\boldsymbol{v}_t$, then the distribution $\mu_t$ evolves according to the continuity equation $\partial_t \mu_t(\boldsymbol{x}) = -\nabla \cdot [\mu_t(\boldsymbol{x}) \boldsymbol{v}_t(\boldsymbol{x})]$, $\boldsymbol{x} \in \mathbb{R}^m, t \geq 0$, where $\nabla \cdot$ denotes the divergence operator.*

**Theorem 7** *(Optimal transport map) The global minimum $f_t^*$ of $\mathcal{L}(f)$ is attained at*

$$f_t^*(\widetilde{\boldsymbol{x}}) = \widetilde{\boldsymbol{x}} - \frac{1}{\nu_t * \mu_0(\widetilde{\boldsymbol{x}})} \int_{\mathbb{R}^m} \varepsilon \nu_t(\varepsilon) \mu_0(\widetilde{\boldsymbol{x}} - \varepsilon) d\varepsilon,$$

*where $*$ denotes the convolution operator.*

**Proof**   The proof follows from the calculus of variations.

$$\mathcal{L}(f) = \int_{\mathbb{R}^m} \mathbb{E}_{\boldsymbol{\varepsilon}} \left[ \|f(\boldsymbol{x} + \boldsymbol{\varepsilon}) - f'(\boldsymbol{x})\|^2 \right] \mu_0(\boldsymbol{x}) d\boldsymbol{x}$$

$$= \int_{\mathbb{R}^m} \mathbb{E}_{\boldsymbol{\varepsilon}} \left[ \|f(\boldsymbol{x}') - f'(\boldsymbol{x}' - \boldsymbol{\varepsilon})\|^2 \mu_0(\boldsymbol{x}' - \boldsymbol{\varepsilon}) \right] d\boldsymbol{x}',$$

where the second line holds by the calculus of variations, *i.e.*, $\boldsymbol{x}' = \boldsymbol{x} + \boldsymbol{\varepsilon}$. Then, for an arbitrary function $h$, the first variation $\delta\mathcal{L}(h)$ is given by

$$\delta\mathcal{L}(h) = \frac{d}{ds} \mathcal{L}(f + sh) \bigg|_{s=0}$$

$$= \int_{\mathbb{R}^m} \frac{\partial}{\partial s} \mathbb{E}_{\boldsymbol{\varepsilon}} \left[ \|f(\boldsymbol{x}) + sh(\boldsymbol{x}) - f'(\boldsymbol{x} - \boldsymbol{\varepsilon})\|^2 \mu_0(\boldsymbol{x} - \boldsymbol{\varepsilon}) \right] d\boldsymbol{x} \bigg|_{s=0}$$

$$= 2 \int_{\mathbb{R}^m} \mathbb{E}_{\boldsymbol{\varepsilon}} \left[ (f(\boldsymbol{x}) - f'(\boldsymbol{x} - \boldsymbol{\varepsilon})) \mu_0(\boldsymbol{x} - \boldsymbol{\varepsilon}) \right] h(\boldsymbol{x}) d\boldsymbol{x}.$$

At a critical point $f^*$ of $\mathcal{L}$, $\delta\mathcal{L}(h) = 0$ for every $h$. Hence,

$$\mathbb{E}_{\varepsilon}\left[(f(\boldsymbol{x}) - f'(\boldsymbol{x} - \varepsilon))\,\mu_0(\boldsymbol{x} - \varepsilon)\right] = 0, \quad a.e. \ \boldsymbol{x},$$

by the fundamental lemma of calculus of variations for integrable functions, and we have

$$f^*(\boldsymbol{x}) = \frac{\mathbb{E}_{\varepsilon}\left[f'(\boldsymbol{x} - \varepsilon)\mu_0(\boldsymbol{x} - \varepsilon)\right]}{\mathbb{E}_{\varepsilon}\left[\mu_0(\boldsymbol{x} - \varepsilon)\right]}.$$

Note that $f^*$ attains the global minimum, because, for every function $h$,

$$\mathcal{L}(f^* + h) = \int_{\mathbb{R}^m} \mathbb{E}_{\varepsilon}\left[\|f^*(\boldsymbol{x}) - f'(\boldsymbol{x} - \varepsilon) + h(\boldsymbol{x})\|^2 \mu_0(\boldsymbol{x} - \varepsilon)\right] d\boldsymbol{x}$$

$$= \int_{\mathbb{R}^m} \mathbb{E}_{\varepsilon}\left[\|f^*(\boldsymbol{x}) - f'(\boldsymbol{x} - \varepsilon)\|^2 \mu_0(\boldsymbol{x} - \varepsilon)\right] d\boldsymbol{x} + \int_{\mathbb{R}^m} \mathbb{E}_{\varepsilon}\left[\|h(\boldsymbol{x})\|^2 \mu_0(\boldsymbol{x} - \varepsilon)\right] d\boldsymbol{x}$$

$$+ 2\int_{\mathbb{R}^m} \mathbb{E}_{\varepsilon}\left[h(\boldsymbol{x})^{\mathsf{T}}\left(f^*(\boldsymbol{x}) - f'(\boldsymbol{x} - \varepsilon)\right)\mu_0(\boldsymbol{x} - \varepsilon)\right] d\boldsymbol{x}$$

$$= \mathcal{L}(f^*) + \mathcal{L}(h) + 2 \cdot 0 \geq \mathcal{L}(f^*).$$

$\square$

**Theorem 1** *If the target network is a residual unit $f'(\widetilde{\boldsymbol{x}}) = \boldsymbol{x} + \sigma(\widetilde{\boldsymbol{x}})$, where $\sigma$ contains fully connected layer and activation function. When $\nu_t := \mathcal{N}(\boldsymbol{0}, t\boldsymbol{I})$, then the global minimum $f_t^*$ is*

$$f_t^*(\widetilde{\boldsymbol{x}}) = \widetilde{\boldsymbol{x}} + t\nabla \log(\nu_t * \mu_0)(\widetilde{\boldsymbol{x}}) + \sigma(\widetilde{\boldsymbol{x}}) := \widetilde{\boldsymbol{x}} + g_t(\widetilde{\boldsymbol{x}}).$$

**Proof** Using Stein's identity (Liu, 1994), $-t\nabla\nu_t(\varepsilon) = \varepsilon\nu_t(\varepsilon)$, which is known to hold only for Gaussians.

$$f_t^*(\widetilde{\boldsymbol{x}}) = \widetilde{\boldsymbol{x}} - \frac{1}{\nu_t * \mu_0(\widetilde{\boldsymbol{x}})} \int_{\mathbb{R}^m} \varepsilon\nu_t(\varepsilon)\mu_0(\widetilde{\boldsymbol{x}} - \varepsilon)d\varepsilon + \sigma(\widetilde{\boldsymbol{x}})$$

$$= \widetilde{\boldsymbol{x}} + \frac{1}{\nu_t * \mu_0(\widetilde{\boldsymbol{x}})} \int_{\mathbb{R}^m} t\nabla\nu_t(\varepsilon)\mu_0(\widetilde{\boldsymbol{x}} - \varepsilon)d\varepsilon + \sigma(\widetilde{\boldsymbol{x}})$$

$$= \widetilde{\boldsymbol{x}} + \frac{t\nabla\nu_t * \mu_0(\widetilde{\boldsymbol{x}})}{\nu_t * \mu_0(\widetilde{\boldsymbol{x}})} + \sigma(\widetilde{\boldsymbol{x}})$$

$$= \widetilde{\boldsymbol{x}} + t\nabla \log(\nu_t * \mu_0(\widetilde{\boldsymbol{x}})) + \sigma(\widetilde{\boldsymbol{x}}).$$

$\square$

**Theorem 2** *Based on the equivalent condition in Theorem 2 (Belavkin, 2016), and let $\Delta$ be the Laplacian operator. At the initial moment $t \to 0$, the pushforward $f_{\#}\mu_t$ with Gaussian distribution satisfies the backward heat equation (BHE), and evolves according to Wasserstein gradient flow: $\partial_t \mu_{t=0}(\boldsymbol{x}) = -\Delta\mu_0(\boldsymbol{x}) = -\operatorname{grad} \mathcal{W}^2(\mu_0, \mu), \boldsymbol{x} \in \mathbb{R}^m$.*

**Proof** The initial velocity vector is given by

$$\partial_t f_{t=0}^*(\boldsymbol{x}) = \lim_{t \to 0} \frac{f_t^*(\boldsymbol{x}) - \boldsymbol{x}}{t} = \nabla \log \mu_0(\boldsymbol{x}) + \sigma(\boldsymbol{x}). \tag{13}$$

Hence, by substituting the score (13) in the continuity equation of Proposition 2, we have

$$\partial_t \mu_{t=0}(\boldsymbol{x}) = -\nabla \cdot [\mu_0(\boldsymbol{x})(\nabla \log \mu_0(\boldsymbol{x}) + \sigma(\boldsymbol{x}))] = -\Delta\mu_0(\boldsymbol{x}).$$

We leave the proof of $\Delta\mu_0(\boldsymbol{x}) = \mathcal{W}^2(\mu_t, \mu)$ in Theorem 3. $\square$

**Theorem 3** *Based on the equivalent condition in Theorem 2 (Belavkin, 2016), and when the noise distribution is a Gaussian distribution, then the pushforward measure $\mu_t := \varphi_{t\#}\mu_0$ evolves according to Wasserstein gradient flow as follows:*

$$\frac{d}{dt}\mu_t(\boldsymbol{x}) = -\Delta\mu_t(\boldsymbol{x}) = -\operatorname{grad}\mathcal{W}^2(\mu_t, \mu), \quad \mu_{t=0}(\boldsymbol{x}) = \mu_0(\boldsymbol{x}). \tag{14}$$

**Proof**   Let the potential function be equal to Kullback–Leibler divergence be $KL(\mu_t, \mu) = \int \mu_t(\boldsymbol{x}) \log \frac{\mu_t(\boldsymbol{x})}{\mu(\boldsymbol{x})} - \mu_t(\boldsymbol{x}) + \mu(\boldsymbol{x}) d\boldsymbol{x}$, then $V_t = -(\log(\mu_t) - \log(\mu))$, then

$$\mathrm{grad} F(\mu_t) = \nabla \cdot (\mu_t \nabla (\log(\mu_t) - \log(\mu))) = \Delta \mu_t.$$

The above continuity also satisfies the case of Wasserstein distance, then using the equivalent condition in Theorem 2 (Belavkin, 2016), we have

$$\partial_t \mu_{t=0}(\boldsymbol{x}) = -\mathrm{grad}\ \mathcal{W}^2(\mu_t, \mu), \boldsymbol{x} \in \mathbb{R}^m.$$

$\square$

We assume the distribution $\mu_l^{(t)}$ is Gaussian distribution. The $\mathcal{W}$-distance of two Gaussian distribution is defined as follow.

**Proposition 3** *(Gelbrich, 1990) Given two Gaussian distributions $\mu_1 \sim \mathcal{N}(\boldsymbol{u}_1, \boldsymbol{\Sigma}_1)$ and $\mu_2 \sim \mathcal{N}(\boldsymbol{u}_2, \boldsymbol{\Sigma}_2)$, then Wasserstein distance between $\mu_1$ and $\mu_2$ is*

$$\mathcal{W}^2(\mu_1, \mu_2) = \|\boldsymbol{u}_1 - \boldsymbol{u}_2\|^2 + \mathrm{tr}\left(\boldsymbol{\Sigma}_1 + \boldsymbol{\Sigma}_2 - 2\left(\boldsymbol{\Sigma}_1^{\frac{1}{2}} \boldsymbol{\Sigma}_2 \boldsymbol{\Sigma}_1^{\frac{1}{2}}\right)^{\frac{1}{2}}\right).$$

Batch Normalization is proposed to stabilize the distributions of layer distribution by introducing the parameters $\gamma^t$ and $\beta^t$ in the $t$-th optimization step. Let $\boldsymbol{\Gamma}_l^{(t)} = \mathrm{diag}(\boldsymbol{\gamma}_l^{(t)})$ and $\boldsymbol{\Gamma}_l^{(t+1)} = \mathrm{diag}(\boldsymbol{\gamma}_l^{(t+1)})$.

**Proposition 4** *Assume the distributions in two iterations in the $l$-th layer are Gaussian distributions, i.e., $\mu_l^{(t)} \sim \mathcal{N}(\boldsymbol{\beta}_l^{(t)}, \boldsymbol{\Gamma}_l^{(t)})$ and $\mu_l^{(t+1)} \sim \mathcal{N}(\boldsymbol{\beta}_l^{(t+1)}, \boldsymbol{\Gamma}_l^{(t+1)})$, then the $\mathcal{W}$-distance between $\mu_l^{(t)}$ and $\mu_l^{(t+1)}$ is $\mathcal{W}^2(\mu_l^{(t)}, \mu_l^{(t+1)}) = \|\boldsymbol{\beta}_l^{(t)} - \boldsymbol{\beta}_l^{(t+1)}\|^2 + \|(\boldsymbol{\Gamma}_l^{(t)} - \boldsymbol{\Gamma}_l^{(t+1)})^2\|^2$.*

**Proof**   Directly from 3, and let $\boldsymbol{u}_1 = \boldsymbol{\beta}_l^{(t)}, \boldsymbol{u}_2 = \boldsymbol{\beta}_l^{(t+1)}$ and $\boldsymbol{\Sigma}_1 = \boldsymbol{\Gamma}_l^{(t)}, \boldsymbol{\Sigma}_2 = \boldsymbol{\Gamma}_l^{(t+1)}$, we have

$$\mathcal{W}^2(\mu_l^{(t)}, \mu_l^{(t+1)}) = \|\boldsymbol{\beta}_l^{(t)} - \boldsymbol{\beta}_l^{(t+1)}\|^2 + \|(\boldsymbol{\Gamma}_l^{(t)} - \boldsymbol{\Gamma}_l^{(t+1)})^2\|^2.$$

$\square$

# D   OPTIMAL TRANSPORT

**Definition 10  (Optimal Transport (Villani, 2008))** *Given a cost function $c$: $\mathcal{K} \times \mathcal{K} \to \mathbb{R}$, the optimal transport distance measures the optimal plan to transport the mass from a probability measure $\widehat{\mu}$ to another probability measure $\mu$:*

$$\widetilde{\mathcal{W}^2}(\widehat{\mu}, \mu) = \inf_{\pi \in \Pi(\widehat{\mu}, \mu)} \iint_{\mathcal{K} \times \mathcal{K}} c(\widehat{\boldsymbol{\kappa}}, \boldsymbol{\kappa}) \pi(d\widehat{\boldsymbol{\kappa}}, d\boldsymbol{\kappa}), \tag{15}$$

*where $\Pi(\widehat{\mu}, \mu)$ is the set of joint probability measures $\pi$ on $\mathcal{K} \times \mathcal{K}$ with the marginals $\widehat{\mu}$ and $\mu$.*

In this paper, we consider the multi-label classification problem. We first introduce the definition of Wasserstein distance. For the case of probability measures, they are histograms in the simplex $\Delta_K$.

**Definition 11  ($\mathcal{W}$-distance)** *Given an input $\boldsymbol{x} \in \mathcal{X}$ and any $f$: $\mathcal{X} \to \Delta_K$, let $f(\boldsymbol{x})$ be the predicted label distribution $\widehat{\mu}$, and let $\boldsymbol{y}$ be the target distribution $\mu$, then the $\mathcal{W}$-distance between $\widehat{\mu}$ and $\mu$ is*

$$\mathcal{W}^2(\widehat{\mu}, \mu) = \inf_{\boldsymbol{T} \in \Pi(\widehat{\mu}, \mu)} \langle \boldsymbol{T}, \boldsymbol{C} \rangle, \tag{16}$$

*where $\boldsymbol{C} \in \mathbb{R}_+^{K \times K}$ is the cost matrix whose the element can be defined as $C_{\kappa, \kappa'} = d_{\mathcal{K}}^p(\kappa, \kappa')$, and the set of couplings is composed of joint probability distributions with their marginals $\hat{\mu}$ and $\mu$, i.e., $\Pi(\widehat{\mu}, \mu) = \{\boldsymbol{T} \in \mathbb{R}_+^{K \times K} : \boldsymbol{T}\mathbf{1} = \widehat{\mu}, \boldsymbol{T}^\top \mathbf{1} = \mu\}$.*

**Entropic Wasserstein loss.**   Cuturi (2013) introduces an entropic regularization such that the non-convex problem (2) can be turned to a strictly convex problem:

$$\widehat{\mathcal{W}^2}_\alpha(\widehat{\mu}, \mu) := \inf_{\boldsymbol{T} \in \Pi(\widehat{\mu}, \mu)} \langle \boldsymbol{T}, \boldsymbol{C} \rangle - \frac{1}{\alpha} H(\boldsymbol{T}), \tag{17}$$

where $H(\boldsymbol{T}) = -\sum_{\kappa, \kappa'} T_{\kappa, \kappa'}(\log(T_{\kappa, \kappa'}) - 1)$.

With the help of the entropic regularization, we solve optimal transport problem efficiently using Sinkhorn-Knopp matrix scaling algorithm (Knight, 2008).

**Proposition 5** *The transport matrix $\boldsymbol{T}^*$ optimizing Problem (17) satisfies $\boldsymbol{T}^* = \text{diag}(\boldsymbol{u})\boldsymbol{K}\text{diag}(\boldsymbol{v})$, where $\boldsymbol{K} = e^{-\alpha \boldsymbol{M}}$, $\boldsymbol{u} = e^{\alpha \boldsymbol{a}}$ and $\boldsymbol{v} = e^{\alpha \boldsymbol{b}}$, where $\boldsymbol{a}$ and $\boldsymbol{b}$ are the Lagrange dual variables.*

**Theorem 8** *(Genevay et al., 2017)Wasserstein sinkhorn loss between the predicted label distribution $\widehat{\mu}$ and the target label distribution $\mu$ is defined as:*

$$\bar{\mathcal{W}}^2(\widehat{\mu}, \mu) := 2\widehat{\mathcal{W}^2}_\alpha(\widehat{\mu}, \mu) - \widehat{\mathcal{W}^2}_\alpha(\widehat{\mu}, \widehat{\mu}) - \widehat{\mathcal{W}^2}_\alpha(\mu, \mu), \tag{18}$$

*with the following limiting behavior as $\alpha \to 0$: $\bar{\mathcal{W}}^2(\widehat{\mu}, \mu) \to 2\widehat{\mathcal{W}^2}_\alpha(\widehat{\mu}, \mu)$.*

Note that normalized Wasserstein loss is non-negative and $\bar{\mathcal{W}}^2_\alpha(\widehat{\mu}, \mu) = 0$ if and only is $\widehat{\mu} = \mu$. In the quantification, we use normalized Wasserstein loss to measure the divergence between distributions.

# E INTERMEDIATE LAYER

We add auxiliary classifiers for the intermediate layers as mentioned below. For ResNet-18, we truncate each block and add a classifier after its feature map. We label these intermediate layers from 0-th layer to 8-th layer. The truncation of ResNet-50 is analogous to ResNet-18. For VGG-16, we add a classifier after each convolution layer and label them from 0-th layer to 12-th layer.

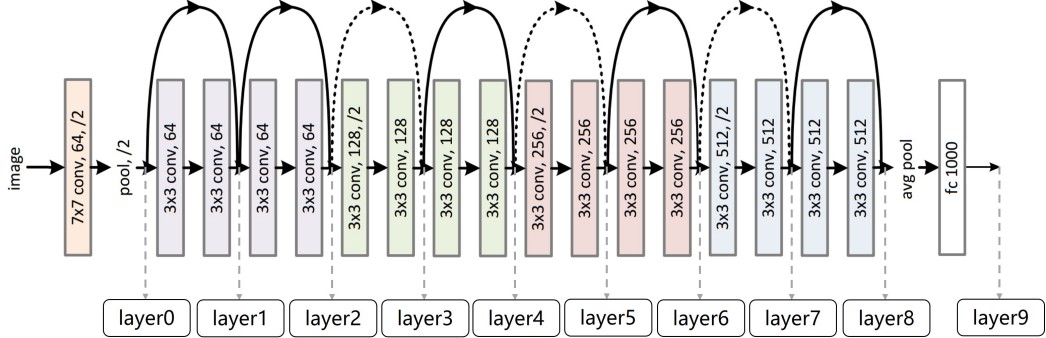

Figure 7: Definition of intermediate layer of ResNet-18.

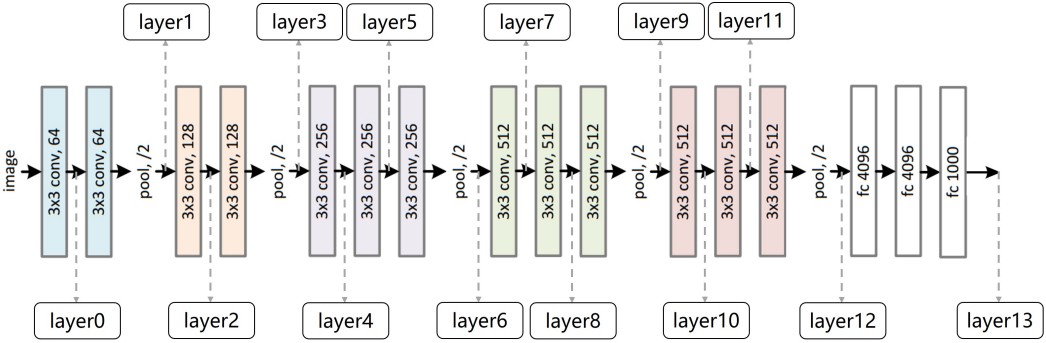

Figure 8: Definition of intermediate layer of VGG-16.

# F ACROSS-LAYER DISTRIBUTION PROPAGATION

We show more results for different datasets, measures and networks across different layers.

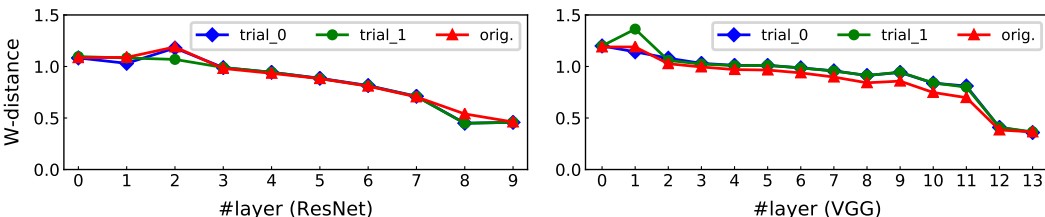

Figure 9: Wasserstein distance across different layers of ResNet-18 (left) and VGG-16 (right) on different test set. Specifically, we shuffle the VOC2007 dataset (including training set and test set) and divide it into training set and test set following the ratio of original dataset. We shuffle twice and define them as "trial_0" and "trial_1", respectively. Besides, the "orig." means that we use the original dataset. From Figure 9, different experiments consistently have the same decreasing tendency through the depth of a neural network.

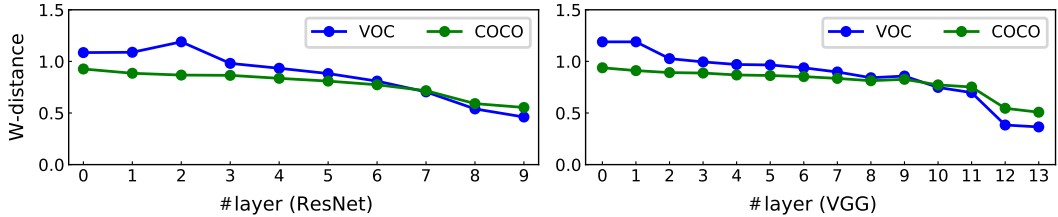

Figure 10: Wasserstein distance across different layers of ResNet-18 (left) and VGG-16 (right) on test set. The tendency of distribution propagation on test set is the same as the training set(Figure 3), suggesting that the distribution propagation is irrelevant to the property of the dataset.

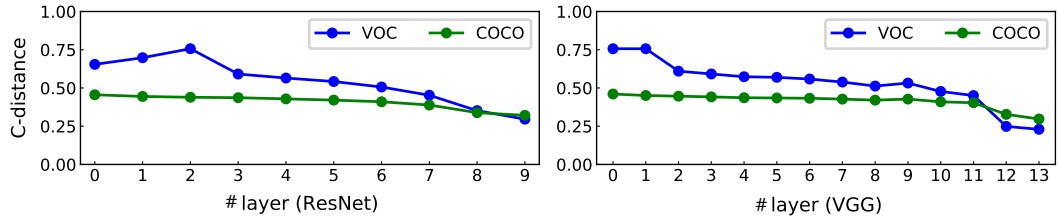

Figure 11: Chebyshev distance across different layers of ResNet-18 (left) and VGG-16 (right) on training set. Although the tendency of Chebyshev distance is analogous to Wasserstein distance, Chebyshev distance ignores any metric structure (Frogner et al., 2015). Therefore, we use Wasserstein distance to quantify the discrepancy of label distributions.

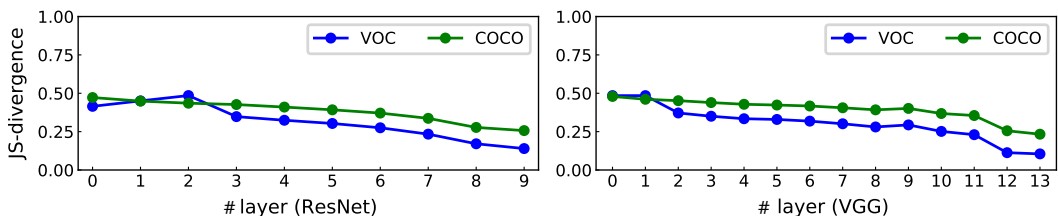

Figure 12: Jensen–Shannon divergence across different layers of ResNet-18 (left) and VGG-16 (right) on training set. The conclusion of Jensen–Shannon divergence is the same as the Chebyshev distance.

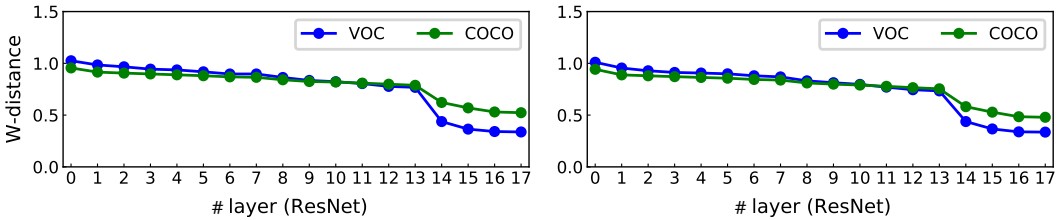

Figure 13: Wasserstein distance across different layers of ResNet-50 on training set (left) and test set (right). We get the same conclusion on ResNet-50 that Wasserstein distance between the distribution of any layer and the target distribution tends to decreases along the depth of a DNN.

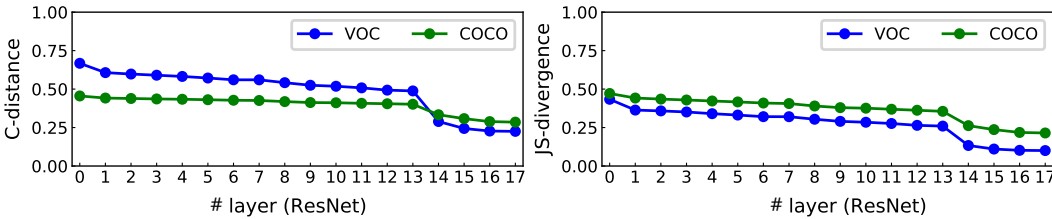

Figure 14: Chebyshev distance (left) and Jensen–Shannon divergence (right) across different layers of ResNet-50 on training set.

# G SINGLE-LAYER DISTRIBUTION PROPAGATION

We present more results about the single-layer distribution propagation when training DNNs. We calculate the Wasserstein distance between the prediction distribution of a selected epoch $i$ and the next epoch $i + 1$ and show in Figure 15. Analogous to Figure 4, Figure 16 shows that the label distribution of one sample propagates from the first epoch to the last epoch.

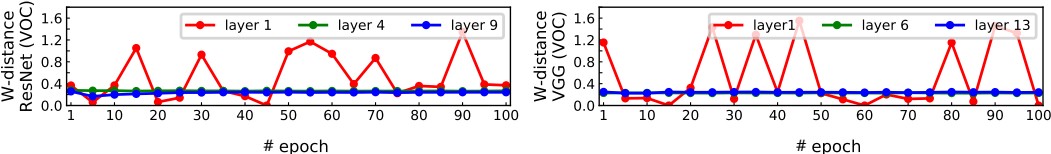

Figure 15: Wasserstein distance between distributions of adjacent training epoch.

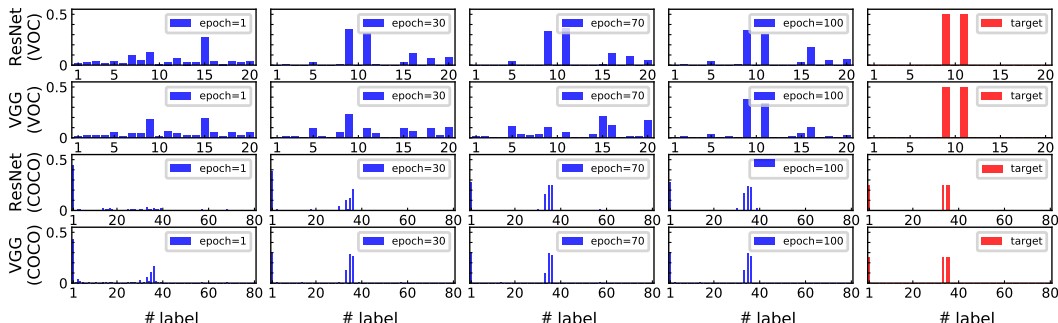

Figure 16: Distribution propagation across different training epochs of ResNet-18 and VGG-16.

## H    EARLY-EXITS STRATEGY

Deep neural networks often occur the over-thinking issue, that is, the samples are correctly classified in the intermediate layer but misclassified in the last layer. We call these samples as confused samples. From Table 2 and 3, the accuracy of ResNet-18 (VOC) and VGG-16 (VOC) are 64.48% and 68.96%, respectively. In other words, out of 35.52% and 31.04% of the samples for ResNet-18 and VGG-16 are misclassified on the test set of VOC2007, respectively. For these misclassified samples, 1.17% are actually correctly classified at 0-th layer, 0.44% are at 1-th layer and so on. Ideally, if we early exit these confused samples, the cumulative accuracy of ResNet-18 is 70.44% on VOC dataset.

Table 2: Over-thinking results of ResNet-18.

| layer | 0 | 1 | 2 | 3 | 4 | 5 | 6 | 7 | 8 | 9 | ideal result |
|---|---|---|---|---|---|---|---|---|---|---|---|
| VOC | 1.17% | 0.44% | 0.69% | 0.71% | 0.55% | 0.50% | 0.53% | 0.85% | 0.53% | 64.48% | **70.44%** |
| COCO | 0.49% | 0.43% | 0.47% | 0.28% | 0.30% | 0.31% | 0.37% | 0.78% | 0.50% | 27.33% | **31.26%** |

Table 3: Over-thinking results of VGG-16.

| layer | 0 | 1 | 2 | 3 | 4 | 5 | 6 | 7 | 8 | 9 | 10 | 11 | 12 | 13 | ideal result |
|---|---|---|---|---|---|---|---|---|---|---|---|---|---|---|---|
| VOC | 2.34% | 0.00% | 0.71% | 0.67% | 0.50% | 0.24% | 0.48% | 0.22% | 0.20% | 0.14% | 0.28% | 0.34% | 1.62% | 68.96% | **76.72%** |
| COCO | 0.35% | 0.35% | 0.27% | 0.38% | 0.33% | 0.19% | 0.21% | 0.17% | 0.18% | 0.13% | 0.38% | 0.37% | 1.63% | 32.41% | **37.35%** |

**Early-exits strategy.** We propose a simple probability mechanism to exit confused samples. Specifically, we add an auxiliary classifier for a selected intermediate layer to get the prediction probability distribution. Each auxiliary classifier contains fully connected layer and Sigmoid function. For each sample, we denote the number of exceeding the threshold $p = 0.5$ as $N$. We also denote the number of exceeding another threshold $q$ as $n$. We denote the ratio as $\gamma$ and $\gamma = n/N$. The threshold $q$ and ratio $\gamma$ are range from 0.5 to 1. We search for the best values of them to improve classification performance and show the results on Table 4.

Table 4: Improve performance of ResNet and VGG.

| Method | VOC | | | | COCO | | | |
|---|---|---|---|---|---|---|---|---|
| | accuracy | CF1 | OF1 | mAP | accuracy | CF1 | OF1 | mAP |
| ResNet-18 | 64.48 | 58.02 | **59.10** | 85.18 | 27.33 | 55.65 | 60.30 | 64.36 |
| ResNet-18+EarlyExits | **66.01** | **58.67** | 58.76 | **85.49** | **30.03** | **57.49** | **61.38** | **67.28** |
| VGG-16 | 68.96 | 58.58 | 59.71 | 88.48 | 32.41 | 59.37 | 62.94 | 70.64 |
| VGG-16+EarlyExits | **69.85** | **59.49** | **59.94** | **88.57** | **33.95** | **60.32** | **63.73** | **71.95** |
| ResNet-50 | 69.79 | 60.93 | **60.23** | 88.84 | 33.81 | 61.25 | 64.01 | 71.77 |
| ResNet-50+EarlyExits | **70.98** | **62.84** | 60.21 | **89.14** | **35.70** | **62.39** | **64.76** | **73.68** |

# I  MORE QUANTITATIVE RESULTS

We demonstrate more results on easy, hard and confused samples. For each sample, we show the image in the upper left, distribution corresponding to the selected layer in the upper right and the distribution propagation across different layers in the bottom.

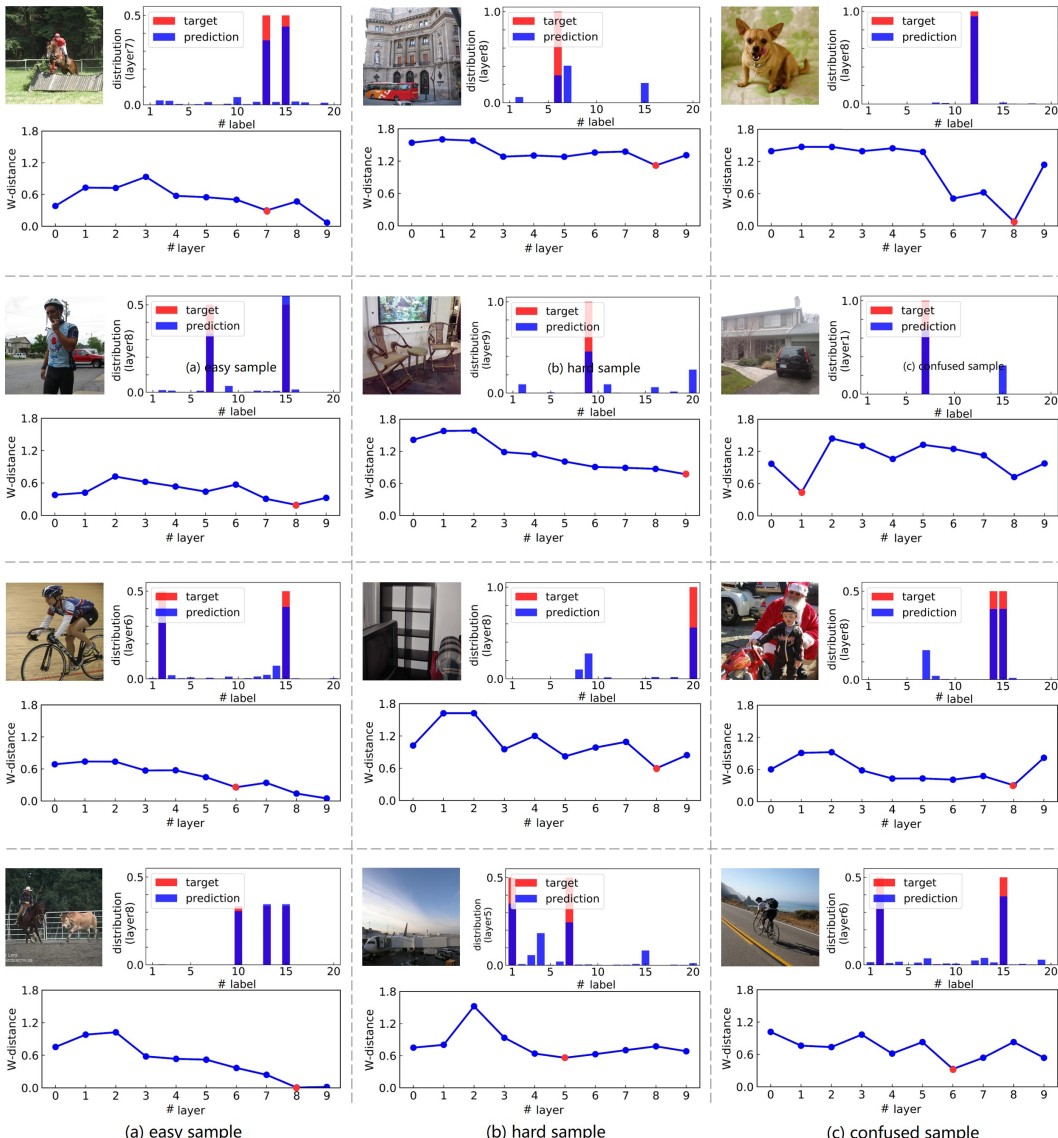

Figure 17: More results of easy, hard and confused samples.

