# OpenReview forum: "Towards Interpreting Deep Neural Networks via Understanding Layer Behaviors"
_ICLR.cc/2020/Conference — Reject_

### Official Review · AnonReviewer3 · 2019-10-22
**Official Blind Review #3**

**Rating:** 3

**Review:**

This paper seeks to understand both across-layer and single-layer behavior within neural networks (i.e. layer behavior along the depth of a network, and behavior of a single layer along training epochs). Therefore, they resort to the optimal transport framework to compare predicted and target distributions. Theoretically, they show that the Wasserstein distance between predicted and target distributions is decreasing along the depth and for a single layer, along training iterations. They also give intuition on how this analysis can help the learning process in practice.

This paper gives an interesting contribution to the in-depth analysis of neural networks. However, some elements remain unclear:

1.	The setting of multi-label classification does not really motivate the use of measures.
2.	It is unclear why the use of teacher/student networks are pertinent or necessary.
3.	There is no detail on the regularization strength of the Wasserstein distance, or what p (in definition 1) is chosen either in the experiments or in the theorems.
4.	I believe it is understated that all \tilde{f}_i have the same input and output domains (as well as h=h_i in figure 1a), which is restrictive and should have been made clearer.

- Post rebuttal: I thank the authors for their response. On this basis, I am maintaining my weak reject rating.

**Experience Assessment:**

I have published one or two papers in this area.

**Review Assessment: Checking Correctness Of Derivations And Theory:**

I assessed the sensibility of the derivations and theory.

**Review Assessment: Checking Correctness Of Experiments:**

I assessed the sensibility of the experiments.

**Review Assessment: Thoroughness In Paper Reading:**

I read the paper at least twice and used my best judgement in assessing the paper.

---

> ### Author Response · Authors · 2019-11-14
> **Response to Reviewer #3**
>
> Thank you for your valuable comments. We have carefully considered the four concerns and made the paper clearer in the revised paper. We sincerely hope you would be satisfied with the clarifications below.
>
> Q1. Concern on multi-label classification and Wasserstein distance
>
> The setting of multi-label classification does motivate the use of Wasserstein distance. First, using Wasserstein distance is able to improve the performance of multi-label classification [5, 6]. Second, deep neural networks on the multi-label classification task still lack strong theoretical understanding. With the help of optimal transport theory, we are able to use Wasserstein distance to interpret deep neural networks via understanding layer behaviors.
>
> Q2. Necessity of teacher-student networks
>
> We exploit the teacher-student framework to build our analysis as it is a flexible framework to analyze and understand neural networks [7, 8, 9]. Specifically, in our one-layer behavior analysis, this framework helps to understand the dynamics of a student network from the teacher network. In our multi-layer behavior analysis, this framework helps to study the ability of the student network to express distributions of the teacher network (Barron function). Therefore, the teacher-student analysis framework is important and necessary in our paper to understand deep neural networks. We believe our analysis framework would provide a different view of understanding and interpreting neural networks.
>
> Q3.	More details of the regularization strength and p of Wasserstein distance
>
> Regularization strength of the Wasserstein distance: When the regularization strength $\alpha$ is large enough, the entropic Wasserstein distance in Eqn. (17) coincides with the Wasserstein distance in Eqn. (2) [10]. In practice, we set $\alpha=0.01$ to achieve balanced results. In addition, we choose $p=2$ in the experiments and all theoretical analysis.
>
> Q4.	More details of $\tilde{f}_i, f_i$ and their domains
>
> In the second paragraph of Section 3, for all functions $\tilde{f}_i, i=1, …, L$, they have different input and output domains. In contrast, for all functions $f_i, i=1, …, L$, they have the same input domain and the same output domain, because it feeds the same input and then outputs the label distribution to close to the ground-truth. We clarify Figure 1 (a) and make them clearer in the revised paper.
>
>
> Reference:
> [5] Charlie Frogner et al. Learning with a wasserstein loss. NeurIPS, 2015.
> [6] Peng Zhao et al. Label distribution learning by optimal transport. IJCAI. 2018.
> [7] Yuandong Tian. An analytical formula of population gradient for two-layered ReLU network and its applications in convergence and critical point analysis. ICML, 2016.
> [8] Simon S. Du et al. When is a convolutional filter easy to learn? ICLR, 2018.
> [9] Qiuyi Zhang et al. Electron-proton dynamics in deep learning. arxiv, 2017.
> [10] Marco Cuturi. Sinkhorn Distances: lightspeed computation of optimal transport, NeurIPS, 2013.

---

### Official Review · AnonReviewer1 · 2019-10-23
**Official Blind Review #1**

**Rating:** 3

**Review:**

The authors intuitively, and then analytically, explain the behavior in the hidden layers of deep convolutional networks and show how the behavior can be used to improve performance by "early exiting."

I give this paper a weak reject. I believe this paper does well by connecting the intuitive explanation with the proofs, and then by confirming their results through experimentation. I also applaud the authors for their rigorous explanation of the hyper-parameters and experimentation methods. However, from what I can tell, there was no cross-fold validation or even repeat trials with different partitioning to see whether the differences in performance were just random perturbations or a consistent effect. The increase in accuracy isn't large enough across experiments to allay my concerns.

I think the authors have some very compelling work here, but the lack of a large difference in accuracy combined with insufficient testing methodology causes me to reject this paper... but only barely. I can be convinced otherwise with a compelling set of arguments.

**Experience Assessment:**

I have read many papers in this area.

**Review Assessment: Checking Correctness Of Derivations And Theory:**

I assessed the sensibility of the derivations and theory.

**Review Assessment: Checking Correctness Of Experiments:**

I assessed the sensibility of the experiments.

**Review Assessment: Thoroughness In Paper Reading:**

I read the paper at least twice and used my best judgement in assessing the paper.

---

> ### Author Response · Authors · 2019-11-14
> **Response to Reviewer #1**
>
> Thank you for your valuable comments. We conduct thorough repeated experiments in the revised paper, and we sincerely hope you would be satisfied with our following response on your concern over the consistency of experimental results.
>
> Q1. Consistency of experimental results
>
> The experimental results are consistent with repeated experiments, which are shown in Figure 9 in Section F of Supplementary materials. In this experiment, we shuffle the data and then conduct three experiments with different partitioning. From Figure 9, different experiments consistently have the same decreasing tendency through the depth of a neural network.
>
> Here we would like to highlight our main contributions as below:
>
> 1. We propose a unified teacher-student analysis method to explore both across-layer and single-layer behaviors.
> i) Across-layer behaviors: The W-distance between the distribution of any layer and the target distribution decreases along the depth of a DNN.
> ii) Single-layer behaviors: For a specific layer, the W-distance between the distribution in an iteration and the target distribution decreases across the training iterations when introducing a loss in the layer.
> iii) We prove that a deep layer is not always better than a shallow layer for some samples (see Figure 5).
>
> 2. We have provided extensive experiments to justify these findings.

---

### Official Review · AnonReviewer2 · 2019-10-29
**Official Blind Review #2**

**Rating:** 6

**Review:**

This paper presents a method to compute the distance of distribution of two layers in neural networks by using the label distribution mapping (e.g., Frogner et al., 2015). With the tool, authors could see how individual layers could related each other across-layer (along the depth) and single layer (training epoch).

I believe that the contributions of this paper are week in analyzing individual layers across-layer since there are many extensive studies are conducted on information bottleneck methods with mutual information. I believe that those methods are better to analyze the dynamics of learning even without the additional label distribution mapping.

However, authors of this paper presents a way to utilize the label distribution mapping to compare the distance of individual layers when an input image come as shown in Figure 5 which I believe the main contribution of this paper.

The (somehow artificial and ambiguous) term, label distribution is used several places before it is defined. Even in Section 3, the label distribution mapping is not clearly explained except for the description of FC+softmax. Thus, it would be better to clarify the definition. Also, it is not clear that the label distribution reflect the actual distribution of (nodes or feature maps) in a specific layer. It would be good to spend more space and resources (e.g., image and/or running examples) to explain the definition of label distribution.

**Experience Assessment:**

I have published one or two papers in this area.

**Review Assessment: Checking Correctness Of Derivations And Theory:**

I assessed the sensibility of the derivations and theory.

**Review Assessment: Checking Correctness Of Experiments:**

I assessed the sensibility of the experiments.

**Review Assessment: Thoroughness In Paper Reading:**

I read the paper at least twice and used my best judgement in assessing the paper.

---

> ### Author Response · Authors · 2019-11-14
> **Response to Reviewer #2**
>
> Thank you for your constructive comments.
>
> Q1. Advantage of the proposed method over information bottleneck and main contributions of our paper
>
> Existing studies [1, 2] using information bottleneck methods mainly analyze the dynamics of across different layers. However, it is hard for these methods to analyze the dynamics of a specific layer through different iterations. In contrast, our proposed method is able to analyze both single-layer and across-layer behaviors.
>
> We highlight the main contributions as follows:
>
> 1. We propose a unified teacher-student analysis method to explore both across-layer and single-layer behaviors.
> i) Across-layer behaviors: The W-distance between the distribution of any layer and the target distribution decreases along the depth of a DNN.
> ii) Single-layer behaviors: For a specific layer, the W-distance between the distribution in an iteration and the target distribution decreases across the training iterations when introducing a loss in the layer.
> iii) We prove that a deep layer is not always better than a shallow layer for some samples (see Figure 5).
>
> 2. We have provided extensive experiments to justify these findings.
>
>
> Q2. Definition of the label distribution
>
> As defined in [3], the label distribution can be defined as a probability distribution to cover a certain number of labels, representing the degree to which each label describes the instance, as shown in Figure 1 (c) of the revised paper. Because the label distribution is a probability distribution, its sum is equal to 1. The revised paper provides more intuitive examples to explain the definition of label distribution.
>
> Due to the convexity of cross-entropy loss, we derive the optimal label distribution for given features of every layer [4]. In this sense, the label distribution reflects the actual distribution of feature maps in a specific layer.
>
> Reference:
> [1] Naftali Tishby et al. Deep learning and the information bottleneck principle. IEEE ITW, 2015.
> [2] Seojin Bang et al. Explaining a black-box using deep variational information bottleneck approach. arxiv, 2019.
> [3] Xin Geng. Label Distribution Learning. KDD, 2016.
> [4] Guillaume Alain, Yoshua Bengio. Understanding intermediate layers using linear classifier probes. arxiv, 2018.

---

### Public Comment · ~pankaj_gupta1 · 2019-09-27
**Nice work. Include additional Reference**

Please also include the following work in your related works section. It understands neural networks (especially RNNs) in explaining their judgements and semantic accumulation behavior.

Pankaj Gupta, Hinrich Schütze. LISA: Explaining Recurrent Neural Network Judgments via Layer-wIse Semantic Accumulation and Example to Pattern Transformation. In BlackboxNLP@EMNLP 2018.

---

> ### Public Comment · ~Cantona_ViVian1 · 2019-09-27
> **I do not think there is a connection between the two works.**
>
> The submitted paper targets at understanding across-layer behavior during the learning process of DNNs. And the mentioned paper is about interpreting the predictions of RNNs. Why they are related?

---

> > ### Public Comment · ~pankaj_gupta1 · 2019-09-28
> > **Semantic Propagation in hidden layers**
> >
> > Appreciate your question about relatedness. I understand the following connection of this work vs [1], though the latter is explanation-based method to study behavior of neural networks.
> >
> > At test time, [1] studies the layer-wise semantic accumulation behavior, how a semantic is built given a sequence of words and propagated across hidden layers of RNN. Moreover, it detects  and extracts salient textual patterns in building semantics. It is based on simply analyzing the label distribution at each of the hidden layers in a sequence classification task and studies the confidence of RNN for the target class, spanning across recurrent layers and thus, in accumulating semantics across hidden layers.
> >
> > However, this work explores how label distribution propagates from one layer to another in order to understand behaviors of different hidden layers in DNN.

---

> ### Author Response · Authors · 2019-10-03
> **Discussions of the related work**
>
> Thanks for your helpful comments. LISA (Gupta et al, 2018) is a great work to explain deep neural networks by understanding the layer-wise semantic accumulation behavior. We will discuss and cite this work in the related work.

---

### Author Response · Authors · 2019-11-14
**Response to AC and all reviewers**


Dear AC and reviewers,

Thank you very much for your constructive comments. In this paper, we propose a unified teacher-student analysis method to analyze both across-layer and single-layer behaviors of neural networks. Moreover, our theoretical findings help to improve the classification performance of multi-label learning tasks (see Table 1). We believe our results would provide a different view of understanding and interpreting neural networks.

We have updated a revised version of the paper. The changes have been highlighted as follows:

1. We highlight the contributions of our paper on page 2.
2. We discuss some studies using information bottleneck methods in related work.
3. We define the label distribution in Figure 1 (c) and Section 3, and provide more intuitive examples in Figure 16 in Section I of Supplementary materials.
4. We conduct thorough repeated experiments to verify the consistency of performance in Figure 9 in Section F of Supplementary materials.
5. We explain the reasonability and necessity for the setting of multi-label classification in Section 4.
6. We explain the importance and necessity for the teacher-student networks in Section 5.
7. We give more details of $f_i$ and $\tilde{f}_i$, and clarify Figure 1 (a) in the revised paper.

Thank you very much for your consideration.

---

### Decision · Program_Chairs · 2019-12-19

[review text omitted: it was posted to a different submission]